# Demystifying the Economic Growth and CO₂ Nexus in Fujian's Key Industries Based on Decoupling and LMDI Model

Qingquan Jiang [1], Jinhuang Lin [1,*], Qianqian Wei [1,*], Rui Zhang [1] and Hongzhen Fu [2]

1   School of Economics and Management, Xiamen University of Technology, Xiamen 361024, China
2   School of Business, Minnan Normal University, Zhangzhou 363000, China
*   Correspondence: 2121021017@s.xmut.edu.cn (J.L.); 2221021008@s.xmut.edu.cn (Q.W.)

**Abstract:** Faced with peaking carbon emissions and carbon neutrality goals, low-carbon transformation has become an important part of China's current economic construction. Fujian is one of the provinces with the fastest economic development in China and the core area of the 21st Century Maritime Silk Road. Therefore, its low-carbon economic development path is of great significance to China. This study focused on the key carbon emission industries in Fujian Province, using energy and carbon emission data from industrial sectors in Fujian Province from 2005 to 2019 to establish the Tapio decoupling model. Then, we decomposed the carbon emission drivers of each industry using the LMDI decomposition method, and finally analyzed the decoupling efforts made by each carbon emission driver on the basis of the Tapio decoupling model and LMDI decomposition model. The results showed that (1) carbon emissions in Fujian Province were mainly concentrated in the manufacturing industry and the electricity, heat, gas, water production and supply industries; (2) to date, some industries in Fujian Province have achieved the decoupling of carbon emissions, but the decoupling status was not stable; and (3) both energy structure and energy intensity have facilitated increasing decoupling efforts for carbon emissions. Industrial structure has contributed less to decoupling, and population size has not yet to make an impact on decoupling. Therefore, in the future, Fujian Province should increase expenditure on green technology research and development to improve energy efficiency and gradually use renewable energy to replace fossil energy, continue to adjust the industrial structure, and increase the government's supervision on corporate carbon emissions.

**Keywords:** Fujian Province; industry; carbon emissions; decoupling; influencing factors

## 1. Introduction

The early stages of global economic development creates large emissions of greenhouse gases such as carbon dioxide, which contribute significantly to global warming [1]. It is generally believed in academic circles that global warming is caused by countries' vigorous economic development and the use of large amounts of fossil energy in the early stage of development [2–4]. In 2013, the Intergovernmental Panel on Climate Change (IPCC) released a global warming assessment report, which stated that the global warming process was underway and the problem was more severe than expected [5]. If countries did not take effective emission reduction measures, the global temperature would continue to increase at the current rate. By that time, ecosystems would be further unbalanced; species diversity would be greatly impacted; and rising sea levels would mean that many islands and low-altitude cities face the threat of flooding.

China is the largest developing country with the largest total carbon emissions in the world and faces huge international pressure to reduce emissions [6]. Since the reform and opening up in China, its economy has developed rapidly. However, in the past, China's economic growth was at the expense of the environment [7]. The pressing pursuit of economic growth led to the increasing consumption of fossil energy in China, and global warming caused by the massive emission of greenhouse gases has seriously affected

the productivity and life of human beings, threatening our survival and development. In his speech to the 75th General Assembly of the United Nations in September 2020, President Xi Jinping made it clear that China would strive to ensure that its carbon dioxide emissions peaked by 2030 and to achieve its carbon neutrality target by 2060, which reflect China's determination and commitment to address climate change [8]. China is a vast country with different economic development situations in different regions, which means that their optimal low-carbon development paths are also different. It is therefore particularly important to formulate carbon emission reduction policies that are appropriate to each region.

Fujian Province belongs to the southeastern coastal region, and its rapid economic development was accompanied by increasing energy consumption, with the inevitable consequence that a large quantity of greenhouse gases were produced. In August 2016, Fujian Province was listed as one of the first national ecological civilization pilot zones, aimed at creating a green cycle low-carbon development pilot zone [9]. Therefore, the exploration of low-carbon development in Fujian Province will become a benchmark for low-carbon construction in other provinces and regions in China [10]. In addition, the "The 14th Five-Year Plan" of Fujian Province clearly stated: "Fujian strives to reduce carbon dioxide emissions per unit of GDP by 13.5%, reduce the energy intensity by 14% compared with 2020, and gradually promote the carbon peak target by sector and industry in the 14th Five-Year Plan period." Therefore, a thorough and comprehensive study of the relationship between carbon emission and economic development and its influencing factors is of great significance for Fujian Province and even China to formulate corresponding policies to achieve carbon emission reduction goals.

There are many researches on the correlation between economic growth and carbon emission. Decoupling theory is one of the important theories to study the relationship between economic growth and carbon emission. Decoupling of carbon emissions is an idealized process in which the relationship between economic growth and greenhouse gas emissions weakens until it disappears, that is, carbon emissions are gradually reduced on the basis of economic growth [11].

This paper aims to explore the decoupling of carbon emissions and economic development in Fujian Province, and focus on identifying important drivers of carbon emissions at industry level in order to provide advice on the future formulation and improvement of carbon emission reduction policies and a reference for other provinces in China to explore green and low-carbon development paths.

Therefore, this paper applied decoupling and decomposing models on Fujian's industry and contributes to the current research in the following aspects. (i) Most of the existing research on carbon decoupling has been limited to national, provincial or single industries. As far as we know, this paper is rare empirical research to comprehensively analyze the decoupling state between carbon emissions and economic growth at industry level in Fujian. (ii) Tapio decoupling model and LMDI index decomposition method are adopted instead of the traditional econometric model. Compared with traditional measurement methods, Tapio decoupling and LMDI method is more concise in calculation, and can dynamically and real-time reflect the annual contribution of various influencing factors to the change of carbon emissions, making the results more intuitive. (iii) In order to explore more influencing factors of decoupling states, this study further decomposed industrial carbon emissions into five factors: population size, economic growth, industrial structure, energy intensity and energy structure. Besides, we also analyze the potential of each factor for carbon emission reduction in different industries.

This paper took Fujian Province as an example and used Tapio decoupling model to study the decoupling relationship between economic growth and carbon emissions in seven important carbon emission industries from 2005 to 2019. Then, Kaya identity and LMDI index decomposition method were used to decompose industrial carbon emissions into five factors: population size, economic growth, industrial structure, energy intensity and energy structure. The influence of these five factors on carbon emission reduction of various

industries was explored, providing theoretical basis for Fujian Provincial government to formulate industrial carbon emission reduction policies. The remaining paper is arranged as, Section 2 provide detail on literature, Section 3 consist of data collection, description, and methodology, Section 4 presents empirical result, and the discussion, and Section 5 presents concussion and suggestions.

## 2. Literature Review

### 2.1. The Relationship between Economic Growth and Carbon Emissions

Exploring the harmonious relationship between economic growth and carbon emissions to promote sustainable economic development has become a topic of great concern to scholars and society. Therefore, in recent decades, a large number of literatures have focused on the relationship between economic growth and carbon dioxide emissions [12]. Early studies usually adopted environmental Kuznets curve (EKC) and econometric methods to investigate the relationship between economic growth and environmental pollution [13–15]. For example, Chen et al. [16] accurately estimated carbon dioxide emissions at provincial and urban scales in China. Then, based on the STRIPAT model, the EKC was verified using 291 cities in China. Ali et al. [17] investigated the effects of economic growth and fossil energy consumption on carbon dioxide emissions in Pakistan during 1975–2014 by using the auto-regressive distributive lag (ARDL) bound test technique. The results of co-integration confirmed that there was a long-term relationship between variables. The results of short-and long-term dynamics confirmed the inverted U-shaped relationship between economic development and $CO_2$ emissions.

Decoupling analysis is also an important theory to study the relationship between economic growth and carbon emissions. The decoupling of $CO_2$ emissions from economic growth means that $CO_2$ emissions no longer increase with economic growth. Compared with EKC method, decoupling analysis is easier to calculate and can dynamically reveal the real-time relationship of different years. Therefore, some scholars believe that this is a more appropriate method to evaluate the relationship between carbon emissions and economic growth [18,19]. At present, some research results have been achieved in low-carbon economy. At national or regional level, Wang and Su [20] adopted the decoupling theory to explore the trend of global and regional decoupling. The results showed that the decoupling state of developed countries was mostly concentrated in the stable weak decoupling state and changes to the strong decoupling state. Most developing countries showed no clear signs of decoupling. Besides, Simbi et al. [21] studied African countries. Jiao et al. [22] studied China and India.

Many scholars have conducted empirical research on the decoupling relationship between economic growth and carbon emissions at industry level. Engo [23] adopted Tapio decoupling method to study the carbon emissions decoupling of industrial sectors in Morocco, Egypt, Tunisia and Algeria. The research results showed that Tunisia achieved low decoupling, while Morocco and Egypt experienced significant decoupling during the study, and the effect of energy structure was an important factor promoting the decoupling of carbon emissions. Besides, a large number of scholars have also emphasized decoupling from the other industry, such as transportation industry [24], steel industry [25], building industry [26], logistics industry [27] and six high-energy intensive industries [28]. A summary of previous studies on the Carbon Emission Decoupling is presented in Table 1.

**Table 1.** A summary of previous studies on the carbon emission decoupling.

| Literature | Object/Period | Method | Finding |
|---|---|---|---|
| Wang and Su (2020) [20] | 192 countries (2000–2014) | Tapio decoupling and LMDI | stable weak decoupling to developed countries. not show a clear decoupling state to most developing countries. |
| Simbi et al. (2021) [21] | African countries (1984–2014) | gravity model, Tapio decoupling and LMDI | Population and economic growth were primary driving forces of $CO_2$ emissions. |
| Jiao et al. (2022) [22] | China and India (1990–2017) | Tapio decoupling and LMDI | remaining weakly decoupling and switch to a strong decoupling in China. continuous fluctuations in India |
| Engo (2020) [23] | Industry of North African countries (1990–2016) | Tapio decoupling and LMDI | low decoupling to Tunisia, significant decoupling to Morocco and Egypt |
| Tapio (2005) [24] | Transportation industry of EU15 countries (1970–2001) | Tapio decoupling | Mostly weak decoupling |
| Wang and Shao (2019) [25] | Steel industry of Jiangsu, China (2005–2013) | Tapio decoupling | Mostly weak decoupling |
| Li and Jiang (2017) [26] | Building industry of China (2005–2015) | Tapio decoupling and LMDI | Mostly expansive negative decoupling |
| Zhang et al. (2018) [27] | Logistics industry of China (1986–2013) | Tapio decoupling | Mostly expanding negative decoupling |
| Du et al. (2018) [28] | Six high-energy intensive industries of China (2001–2016) | LMDI | Most significant in power industry |

Note: log mean Divisia index (LMDI).

## 2.2. Factors Affecting Carbon Emissions

The research on the factors influencing carbon emissions and identification of the key factors causing the changes in carbon emissions can provide a theoretical basis for the formulation of carbon emission reduction policies. At present, researches on factors influencing carbon emissions mainly focus on population, GDP, industrial structure, energy intensity, energy structure and urbanization [29–33]. Decomposition and regression analysis are the two main research methods used in current research on the factors impacting carbon emissions. The decomposition analysis model mainly includes structural decomposition analysis (SDA) and index decomposition analysis (IDA). Index decomposition analysis also includes Laspeyres index decomposition analysis and log mean Divisia index (LMDI) decomposition analysis [34]. Lan and Malik et al. [35] used the SDA model based on data from the MRIO database to study the factors influencing the change in energy footprint of 186 countries from 1990 to 2010. Based on relevant agricultural carbon emissions (ACE) data in Jilin Province from 1998 to 2018, Guo et al. [36] analyzed the factors driving change in ACE in Jilin Province using the Kaya identity and LMDI models. The research showed that the average annual growth rate of ACE in Jilin Province was 1.13% during 1998–2018, and the growth rate gradually decreased. The three main factors causing the reduction in ACE in Jilin Province were identified as production efficiency, industrial structure and the labor force.

Of the regression analysis methods, econometric analysis method is the most widely used. The IPAT and STIRPAT mathematical models are the most widely used. Chontanawat [37] used the IPAT model for empirical analysis of selected ASEAN countries, and the results showed that population size and economic growth were important factors in the IPAT equation that promoted carbon emissions in ASEAN countries, whereas energy use efficiency improvements effectively reduced carbon emissions. Zhu et al. [38] used the Kaya constant equation in combination with the IPAT model to predict carbon dioxide emissions

in Shanxi Province from 2015 to 2040 under different scenarios as well as the years in which carbon emissions would peak. At present, many scholars use the STIRPAT model, which is an extension of the IPAT model and can be used to analyze the nonproportional impact of human factors on the environment. Fu et al. [39] established a system dynamics model based on the STIRPAT model, which included variables such as population, GDP per capita, industrial structure and energy intensity, and used this to analyze the factors influencing carbon emissions in Wanquan Town, Wenzhou City, China, and predicted the future trend of carbon emissions. Xiong et al. [40] used STIRPAT and the Kaya identity to study the effects of urbanization, mechanization, agricultural production efficiency, agricultural structure and level of agricultural economic development on agricultural carbon emissions in Jiangsu province, and the results showed that urbanization was the main factor promoting agricultural carbon emissions, with each 1% increase in urbanization leading to an increase of 0.2510% in agricultural carbon emissions. A summary of previous studies on the influencing factors of carbon emission is presented in Table 2.

**Table 2.** A summary of previous studies on the influencing factors of carbon emission.

| Author | Object/time period | Methods | Findings |
|---|---|---|---|
| Dietz and Rosa (1997) [29] | 111 countries (1989Q1 to 1989Q4) | IPAT model | POP, GDP$\rightarrow$CO$_2\uparrow$ |
| Abid (2016) [30] | 25 sub-Saharan African countries (1996–2010) | OLS, GMM dynamic panel | GDP, RQ$\rightarrow$CO$_2\uparrow$ PS,GE$\rightarrow$CO$_2\downarrow$ |
| Pata (2017) [31] | Turkey (1974–2013) | ARDL | GDP, EC, URB, IS$\rightarrow$CO$_2\uparrow$ |
| Kahia and Aïssa (2016) [32] | 5 Countries in Middle East and North Africa (1980–2012) | PECM | REC$\rightarrow$CO$_2\downarrow$ |
| Zhang et al. (2020) [33] | 281 cities in China (2006–2016) | DSPM | The optimization of IS, TG$\rightarrow$CO$_2\downarrow$ |
| Cui et al. (2019) [34] | 30 provinces in China (2006–2016) | STIRPAT model and OLS | GDP$\rightarrow$CO$_2\uparrow$ RD$\rightarrow$CO$_2\downarrow$ |
| Lan and Malik et al. (2016) [35] | 186 countries (1990–2010) | SDA model | POP, GDP$\rightarrow$CO$_2\uparrow$ |
| Guo et al. (2021) [36] | Jilin Province in China (1998–2018) | Kaya identity and LMDI model | PE, LF$\rightarrow$CO$_2\uparrow$ |
| Chontanawat (2018) [37] | Selected ASEAN countries (1971–2013) | IPAT model | POP, EG$\rightarrow$CO$_2\uparrow$ EUE$\rightarrow$CO$_2\downarrow$ |
| Zhu et al. (2016) [38] | Shanxi Province in China (2015–2040) | Kaya identity and IPAT model | GDP$\rightarrow$CO2$\uparrow$ |
| Fu et al. (2015) [39] | Wanquan Town, Wenzhou City, China (2000–2011) | STIRPAT model | POP,URB,GDP,IS,EI$\rightarrow$CO2$\uparrow$ |
| Xiong et al. (2010) [40] | Jiangsu province in China (1990–2018) | STIRPAT model and Kaya identity | URB$\rightarrow$CO$_2\uparrow$ |

Notice: "$\uparrow$": positive effect, "$\downarrow$": negative effect. Variables: population size (POP); per capita gross domestic product (GDP); regulatory quality (RQ); political stability (PS); government effectiveness (GE), energy consumption (EC), urbanization rate (URB); renewable energy consumption (REC); technical progress (TG); industrial structure (IS); research and development investment (RD); production efficiency (PE); labor force (LF); economic growth (EG); energy use efficiency (EUE); energy intensity (EI). Methods: Ordinary Least Squares (OLS); generalized method of moments (GMM); auto-regressive distributed lag (ARDL); panel error correction model (PECM); dynamic spatial panel model (DSPM); structural decomposition analysis (SDA).

Reviewing the previous empirical literature, it can be found that researches on economic growth and carbon emissions mainly use EKC curve and econometrics methods, and few use decoupling model. The researches on influencing factors of carbon emissions mainly use regression analysis method. Although the influence of accidental random factors can be fully considered, the results are not intuitive enough. At present, studies on the decoupling of carbon emissions and influencing factors of carbon emissions are mainly focusing on the national, regional and single industry levels, and there are few literatures on the research of multiple industries. Considering the shortcomings of existing studies, this study will go on a further analysis to fill these research gaps.

## 3. Materials and Methods

### 3.1. Study Area

Fujian is located on the southeast coast of China (Figure 1), and it is the transportation hub between the East China and South China Seas. South Asia, West Asia and East Africa can all be reached by sea. The province was the starting point of the Maritime Silk Road and Zheng He's voyages to the West. It is also a marine trade distribution center and an important channel connecting the east and west sides of the Taiwan Strait. In 2015, China committed to support Fujian Province in building the core area of the 21st Century Maritime Silk Road. This is particularly conducive to strengthen cooperation between Fujian, ASEAN and other countries along the Maritime Silk Road and create conditions for sustainable future prosperity. In recent years, with the in-depth implementation of the "Belt and Road" initiative, Fujian's exchanges with other countries along the "Belt and Road" have become increasingly frequent and trade has accelerated.

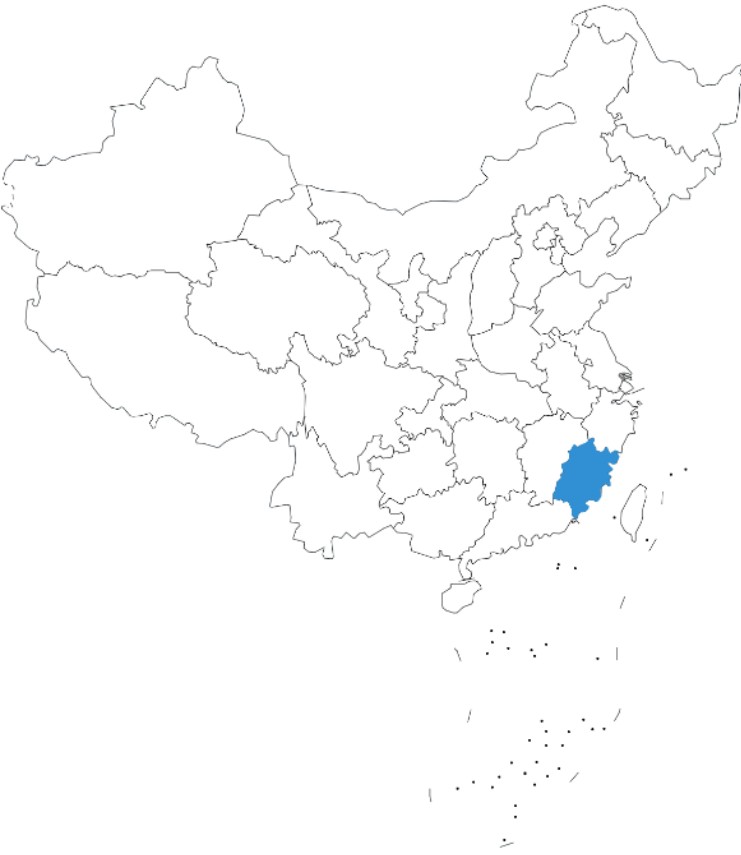

**Figure 1.** Location of Fujian province in China.

According to the China Emission Accounts and Datasets (CEADs), Fujian's total carbon dioxide emissions in 2018 were 261 million tons, ranking 16th in China. Its GDP in 2018 was 3580.404 billion yuan, ranking sixth in China. However, Fujian's GDP in 2021 was 4881.036-billion-yuan, ranking 8th in China. In just three years, the economy grew by nearly 40%. With such rapid economic growth, carbon emission change is an important basis to judge whether Fujian's industrial structure is reasonable, whether its energy consumption and utilization efficiency are appropriate, and whether its economic growth is achieved at the expense of the environment. Fujian Province has entered a golden age of industrialization and rapid economic development, and its economy and society are undergoing major changes. If the industrial structure is unsustainable and energy consumption increases too quickly, Fujian Province will be unable to reach the carbon peak target by 2030 [41]. Therefore, by exploring the decoupling of carbon emissions and economic development

and analyzing the important factors influencing the changes of carbon emissions in Fujian Province, we can provide reference for the sustainable development of Fujian Province and other provinces, and also promote the exchange between Fujian Province and the "Belt and Road" countries in the field of low carbon economy.

### 3.2. Data Resource

The years 2005–2019 were selected as the study period, and the industry energy consumption data and carbon emissions data were secondary data mainly obtained from the China Emission Accounts and Datasets (CEADs). Almost 50 subindustries in the China Carbon Emission Database were divided and categorized into seven major industrial sectors for analysis in accordance with the National Economic Classification of Industries (GB/T 4754–2017). The seven major industrial sectors were: farming, forestry, animal husbandry, fishery and water conservancy (FF); mining (MI); manufacturing (MF); production and supply of electric power, steam and hot water (PE); construction (CI); transportation, storage, post and telecommunication services (TT); and wholesale, retail trade and catering services (WS). There were 17 types of fossil fuel: raw coal, washed coal, other washed coal, coal, coke, coke oven gas, other gas, other coking products, crude oil, gasoline, kerosene, diesel, fuel oil, liquefied petroleum gas, refinery gas, other petroleum products and natural gas. Data relating to gross regional product, gross regional product by industry, resident population number and gross regional product per capita were obtained from the Statistical Yearbook of Fujian Province. To facilitate data comparability, currencies were uniformly converted to 2010 equivalent prices to eliminate the influence of price factors.

### 3.3. Methods

#### 3.3.1. Tapio Decoupling Model

The term "decoupling" originated in the field of physics and was gradually applied in the field of economics. At the end of the 20th century, the Organization for Economic Cooperation and Development (OECD) provided an authoritative interpretation of the word "decoupling": energy consumption generally increases with economic growth, but at a certain stage, the economy continues to grow while energy consumption decreases [42]. Now, scholars at home and abroad are gradually starting to study the decoupling of carbon emissions and economic growth. The Tapio decoupling model is one of the models most used to study the decoupling of carbon emissions and economic growth. Its expression is

$$\varphi_{C,GDP} = \frac{\Delta C / C}{\Delta GDP / GDP} \tag{1}$$

where $\varphi_{C,\,GDP}$ is the elasticity coefficient of decoupling; $\Delta C$ and $\Delta GDP$ respectively represent carbon emission and GDP changes compared with the base period; and C and GDP respectively represent carbon emissions and gross regional product in the base period. The decoupling states can be divided into eight types according to the decoupling elasticity coefficients, as shown in Table 3.

**Table 3.** Decoupling status division table.

| ΔC | ΔGDP | ΦC, GDP | Decoupling Status |
|:---:|:---:|:---:|:---:|
| + | + | (1.2, +∞) | Expansive negative decoupling |
| + | - | (−∞, 0) | Strong negative decoupling |
| - | - | [0, 0.8) | Weak negative decoupling |
| + | + | [0, 0.8) | Weak decoupling |
| - | + | (−∞, 0) | Strong decoupling |
| - | - | (1.2, +∞) | Recessive decoupling |
| + | + | [0.8, 1.2] | Expansive coupling |
| - | - | [0.8, 1.2] | Recessive coupling |

### 3.3.2. Kaya Constant Equation and LMDI Decomposition Model

The Kaya identity was originally proposed in the 1990s by the Japanese scholar Yoichi Kaya for the study of carbon emissions and analysis of their influencing factors [43]. It provides insight into the relationship between carbon dioxide emissions and energy, economy and population in a simple expression:

$$CO_2 = \frac{CO_2}{E} \times \frac{E}{GDP} \times \frac{GDP}{P} \times P \qquad (2)$$

where $CO_2$, E, GDP and P denote the $CO_2$ emissions, energy consumption, gross domestic product and population.

In order to further study the factors influencing industrial carbon emissions in Fujian Province, this study expands and decomposes the original equation on the basis of Kaya identity by adding carbon emission intensity, energy structure and other factors to obtain a new Kaya identity as follows:

$$C = \sum_i \sum_j P \times \frac{G}{P} \times \frac{G_i}{G} \times \frac{E_i}{G_i} \times \frac{E_{ij}}{E_i} \times \frac{C_{ij}}{E_{ij}} = \sum_i \sum_j P \times GP \times IS_i \times EI_i \times ES_{ij} \times f_{ij} \quad (3)$$

where C denote carbon emission; P represents the number of permanent residents in the region; G refers to GDP; $G_i$ refers to GDP of i industry; $E_i$ represents the energy consumption in i industry; $E_{ij}$ represents the j energy consumption in i industry; $C_{ij}$ refers to the j energy carbon emissions in i industry; GP stands for GDP per capita; $IS_i$ represents the proportion of industrial GDP; $EI_i$ represents industrial energy intensity; $ES_{ij}$ represents the proportion of j energy consumption in i industry; and $f_{ij}$ represents the j energy carbon emission coefficient.

The LMDI was proposed by Ang et al. at the end of the 20th century and has been widely used in low-carbon economy research because of its flexible and advanced index factor decomposition technique [44,45]. In our study, the addition and summation decomposition form of the LMDI index decomposition method was used to decompose Equation (3). Because the carbon emission coefficients of various energies are fixed values, $\Delta f = 0$, the results are as follows:

$$\Delta C = C^t - C^0 = \Delta C_P + \Delta C_{GP} + \Delta C_{IS} + \Delta C_{EI} + \Delta C_{ES} \qquad (4)$$

where $\Delta C_P = \sum_i \sum_j \omega_{ij} \times \ln \frac{P^t}{P^0}$; $\Delta C_{GP} = \sum_i \sum_j \omega_{ij} \times \ln \frac{GP^t}{GP^0}$; $\Delta C_{IS} = \sum_i \sum_j \omega_{ij} \times \ln \frac{IS_i^t}{IS_i^0}$; $\Delta C_{EI} = \sum_i \sum_j \omega_{ij} \times \ln \frac{EI_i^t}{EI_i^0}$; $\Delta C_{ES} = \sum_i \sum_j \omega_{ij} \times \ln \frac{ES_{ij}^t}{ES_{ij}^0}$. $\omega_{ij}$ is the defined weight function, which takes the following values: $\omega_{ij} = \begin{cases} \frac{C_{ij}^t - C_{ij}^0}{\ln C_{ij}^t - \ln C_{ij}^0}, C_{ij}^t \neq C_{ij}^0 \\ C_{ij}^t \text{or} C_{ij}^0, C_{ij}^t = C_{ij}^0 \end{cases}$.

Because the expression of weight function loses its meaning when the denominator or logarithm is equal to 0, this paper refers to the corresponding solutions in the different zero-value situations as summarized by Ang et al., see Table 4 [46].

**Table 4.** Treatment of zero value in the LMDI decomposition method.

| Case | $C_{ij}^0$ | $C_{ij}^t$ | $X_0$ | $X_t$ | $\frac{C_{ij}^t - C_{ij}^0}{\ln C_{ij}^t - \ln C_{ij}^0} \ln \frac{X^t}{X^0}$ |
|------|-----------|-----------|-------|-------|-------------------------------------|
| 1 | 0 | + | 0 | + | $C_{ij}^t$ |
| 2 | + | 0 | + | 0 | $-C_{ij}^0$ |
| 3 | 0 | 0 | 0 | 0 | 0 |
| 4 | + | + | 0 | + | 0 |
| 5 | + | + | + | 0 | 0 |
| 6 | + | + | 0 | 0 | 0 |
| 7 | + | 0 | 0 | 0 | 0 |
| 8 | 0 | + | + | + | 0 |

### 3.3.3. Decoupling Effort Model

Decoupling effort refers to the effort made in the industrial development process to directly or indirectly decrease the industrial carbon emissions without harming economic development. In the total carbon emissions, the decoupling effort value of the industry can be obtained by excluding carbon emissions caused by the economic growth effect. According to the LMDI carbon emissions decomposition model, the expression of the industrial carbon emissions decoupling effort indicator D is

$$\Delta E = \Delta C - \Delta C_{GI} = \Delta C_P + \Delta C_{IS} + \Delta C_{EI} + \Delta C_{ES} \tag{5}$$

$$D = -\frac{\Delta E}{\Delta G} \times \frac{G_0}{C_0} = -\left(\frac{\Delta C_P}{\Delta G} - \frac{\Delta C_{IS}}{\Delta G} - \frac{\Delta C_{EI}}{\Delta G} - \frac{\Delta C_{ES}}{\Delta G}\right) \times \frac{G_0}{C_0} = D_P + D_{IS} + D_{EI} + D_{ES} \tag{6}$$

where $\Delta E$ is the sum of each effect of carbon emissions after excluding the economic growth effect; D denotes the decoupling effort indicator after excluding the economic growth effect; and $D_P$, $D_{IS}$, $D_{EI}$, and $D_{ES}$ denote the degree of decoupling effort of carbon emissions caused by changes in population size, industrial structure, energy structure and energy intensity, respectively. When the change in carbon emissions caused by decoupling effort is greater than or equal to zero, i.e., $D \leq 0$, this indicates a state of no decoupling effort. When the change in carbon emissions caused by decoupling effort is less than zero, $0 < D < 1$ indicates a state of weak decoupling effort, and $D \geq 1$ indicates a state of strong decoupling effort.

## 4. Results and Discussion

### 4.1. Current Situation of Carbon Emissions and Decoupling of Industries in Fujian Province

According to the CEADs, changes in industrial carbon emissions in Fujian Province are shown in Figure 2, which demonstrates that overall emissions more than doubled in the 15 years 2005–2019, from 123.90 Mt in 2005 to 278.11 Mt in 2019. The carbon emissions were mainly concentrated in the manufacturing and electricity, heat, gas, water production and supply industries, which were the two major high-energy-consuming industries. From 2005 to 2010, carbon emissions in Fujian Province increased year on year, a result of the early efforts to promote economic growth in the province through vigorous industrial development. However, in the past decade, Fujian's carbon emissions have gradually leveled off. This is similar to the study of China by Guan et al. [47] and Li et al. [48], which may be caused by the improvement of energy efficiency and the optimization of industrial structure in Fujian Province and China in recent years. In the early stage of industrialization, most regions and countries used fossil energy in large quantities in order to vigorously develop their economy. With the economic development reaching a certain level, regions and countries will pay more attention to environmental protection, improve energy efficiency and carry out industrial transformation. The trend of carbon emissions in Fujian Province is also consistent with the EKC hypothesis.

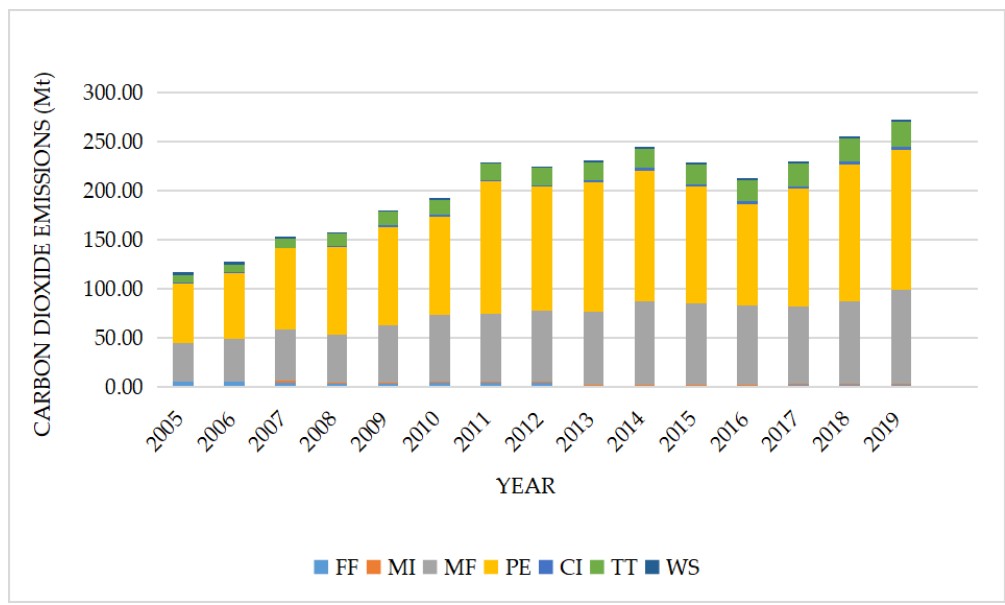

**Figure 2.** 2005–2019 carbon emissions from industries in Fujian Province. Note: abbreviations: farming, forestry, animal husbandry, fishery and water conservancy (FF); mining (MI); manufacturing (MF); production and supply of electric power, steam and hot water (PE); construction (CI); transportation, storage, post and telecommunication services (TI); and wholesale, retail trade and catering services (WS).

The situations for each industry in Fujian Province from 2005 to 2019 with regard to decoupling of economic growth and carbon emissions are shown in Table 5. Of these, the farming, forestry, animal husbandry, fishery and water conservancy industry largely showed strong or weak decoupling. This result is consistent with Han et al. [49]. The agricultural decoupling in southeast coastal areas of China such as Fujian and Guangdong are remarkable. But in recent years the incidence of weak decoupling has increased, indicating that the more recent GDP growth of this industry has caused the growth rate of carbon emissions to increase. Although the industry is still in the decoupling state, attention needs to be paid to improving quality and efficiency. In the future, Fujian Province should not only pay attention to carbon emissions caused by agricultural energy use, but also reduce environmental pollution caused by the use of chemical fertilizers, pesticides and improper treatment of crops such as straw.

The wholesale, retail, accommodation and catering industries were mainly in a strong decoupling state before 2015, but in recent years there has been a trend of change from a strong decoupling state to a weak decoupling state, indicating that the recent development of this industry in Fujian Province may have neglected the protection of the environment. Therefore, Fujian Province should increase efforts to save energy and reduce emissions to sustain the development of the industry while maintaining the previous strong decoupling state.

The construction industry was generally in the weak decoupling state, with a few years of expansive negative decoupling, indicating that the growth rate of carbon emissions in the construction industry was lower than the growth rate of GDP in most years, but the degree of decoupling was not high. This is similar to the study of China by Dong et al. [50]. And their research shows that China can achieve complete decoupling of the construction sector in the future by simply strengthening supervision.

**Table 5.** Decoupling state of economic growth and carbon emissions of industries in Fujian Province.

| Period | FF | MI | MF | PE | CI | TT | WS |
|---|---|---|---|---|---|---|---|
| 2005–2006 | Strong decoupling | Expansive negative decoupling | Expansive coupling | Weak decoupling | Weak decoupling | Weak decoupling | Strong decoupling |
| 2006–2007 | Strong decoupling | Expansive negative decoupling | Expansive negative decoupling | Strong negative decoupling | Strong decoupling | Expansive negative decoupling | Weak decoupling |
| 2007–2008 | Strong decoupling | Strong decoupling | Strong decoupling | Strong negative decoupling | Expansive negative decoupling | Expansive negative decoupling | Expansive negative decoupling |
| 2008–2009 | Expansive negative decoupling | Strong decoupling | Expansive negative decoupling | Expansive negative decoupling | Weak decoupling | Expansive negative decoupling | Strong decoupling |
| 2009–2010 | Expansive coupling | Expansive negative decoupling | Expansive coupling | Weak decoupling | Weak decoupling | Expansive coupling | Expansive coupling |
| 2010–2011 | Weak decoupling | Strong decoupling | Weak decoupling | Strong negative decoupling | Weak decoupling | Expansive negative decoupling | Strong decoupling |
| 2011–2012 | Weak decoupling | Recessive coupling | Weak decoupling | Strong decoupling | Strong decoupling | Weak decoupling | Strong decoupling |
| 2012–2013 | Strong decoupling | Expansive negative decoupling | Weak decoupling | Weak decoupling | Expansive negative decoupling | Weak decoupling | Strong decoupling |
| 2013–2014 | Strong decoupling | Strong decoupling | Expansive negative decoupling | Expansive negative decoupling | Weak decoupling | Weak decoupling | Strong decoupling |
| 2014–2015 | Strong decoupling | Weak negative decoupling | Strong decoupling | Strong decoupling | Weak decoupling | Weak decoupling | Strong decoupling |
| 2015–2016 | Weak decoupling | Strong negative decoupling | Strong decoupling | Recessive coupling | Expansive negative decoupling | Expansive coupling | Strong decoupling |
| 2016–2017 | Strong negative decoupling | Strong decoupling | Strong decoupling | Strong negative decoupling | Expansive negative decoupling | Weak decoupling | Expansive negative decoupling |
| 2017–2018 | Strong decoupling | Weak negative decoupling | Weak decoupling | Expansive negative decoupling | Weak decoupling | Strong negative decoupling | Weak decoupling |
| 2018–2019 | Weak decoupling | Strong decoupling | Expansive coupling | Weak decoupling | Weak decoupling | Weak decoupling | Weak decoupling |

The manufacturing industry was generally moving from expansive negative decoupling and expansive coupling to weak decoupling and then to strong decoupling. This indicates that the development of the manufacturing industry in the past was more dependent on the large consumption of energy, whereas in the past decade, Fujian Province has continued to optimize the energy structure, improve energy utilization and work to achieve the decoupling of economic growth and carbon emissions of this industry. This is similar to the study of Fujian by Yu et al. [51]. Their research shows that Fujian's manufacturing sector has decoupled. Although the decarbonization of Fujian's manufacturing industry is good, the industry is one of the main industries of carbon emissions, and it is still challenging to maintain the decoupling status.

The transportation, storage and postal industries in general showed an encouraging trend from expansive negative decoupling towards weak decoupling, reflecting the effectiveness of energy saving and carbon emission reduction in the transportation industry. However, this industry is also the only industry that has not experienced strong decoupling, and there is still more room for improvement.

The mining industry and electricity, heat, gas and water production and supply industries demonstrated a complicated decoupling state, in particular the electricity, heat, gas and water production and supply industries. Whether in the short or long term, the production and use of electricity have caused serious pollution of the environment, and from the perspective of carbon emissions the industry has had a huge impact [52]. In addition, as an important energy infrastructure industry, electric power is both the energy supply side and the largest energy consumption field. To achieve the goal of carbon neutralization, low carbonization of power supply structure is the key path. So Fujian Province should pay particular attention to the issue of carbon emissions in this industry.

### 4.2. Analysis of Influencing Factors of Industrial Carbon Emissions in Fujian Province

The carbon emission decoupling status of key industries in Fujian Province has been calculated in the previous section. Further, this study used the LMDI to decompose the carbon emissions of key carbon-emitting industries in Fujian Province according to five major factors: population size, economic growth, industry structure, energy intensity and energy structure. The impact of these factors on the carbon emissions of each industry was then analyzed to provide reference opinions for formulating more precise and targeted policies for the achievement of low-carbon development of each industry in Fujian Province in the future. The results of the LMDI decomposition of carbon emissions for each industry are shown in Figure 3a–g.

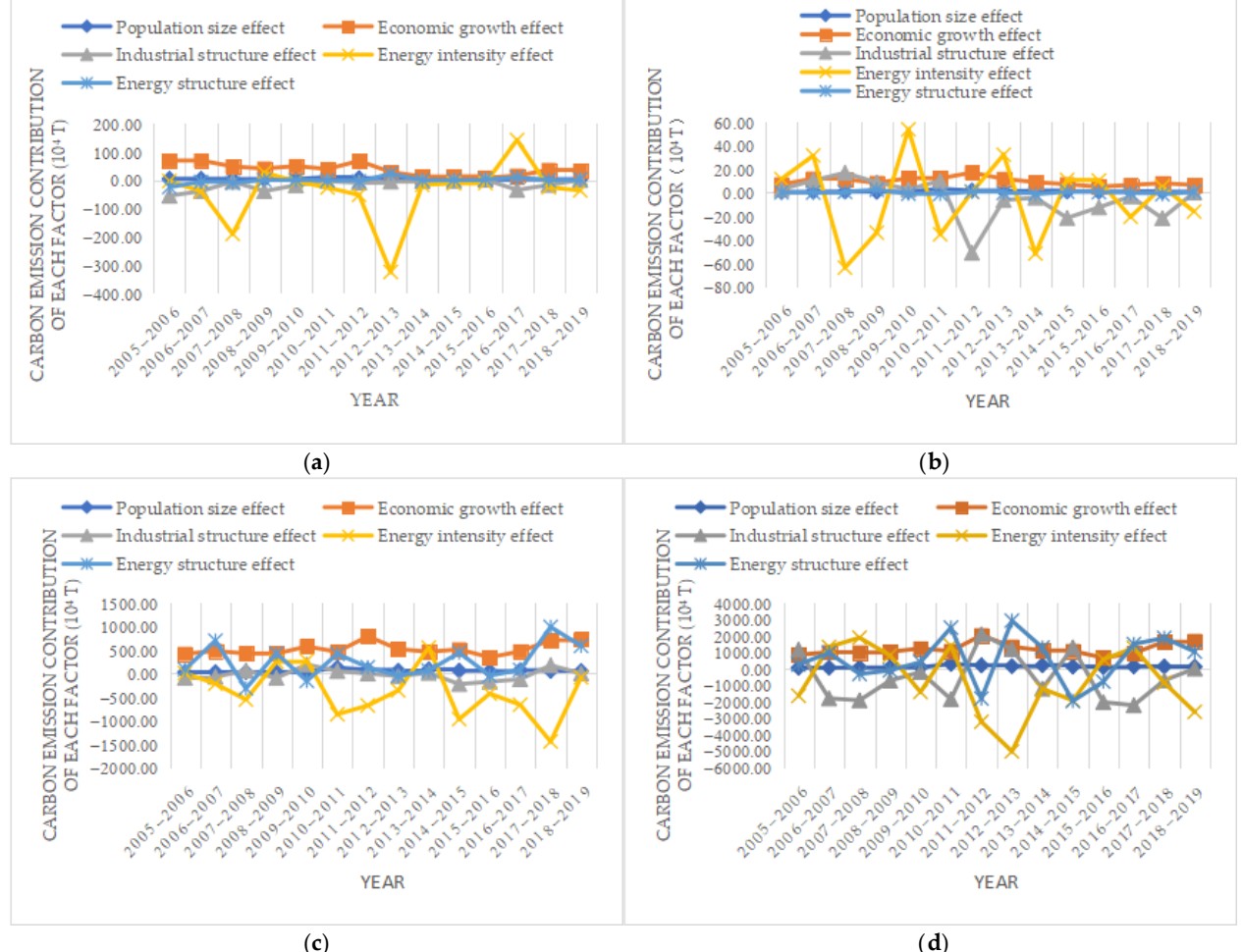

**Figure 3.** *Cont.*

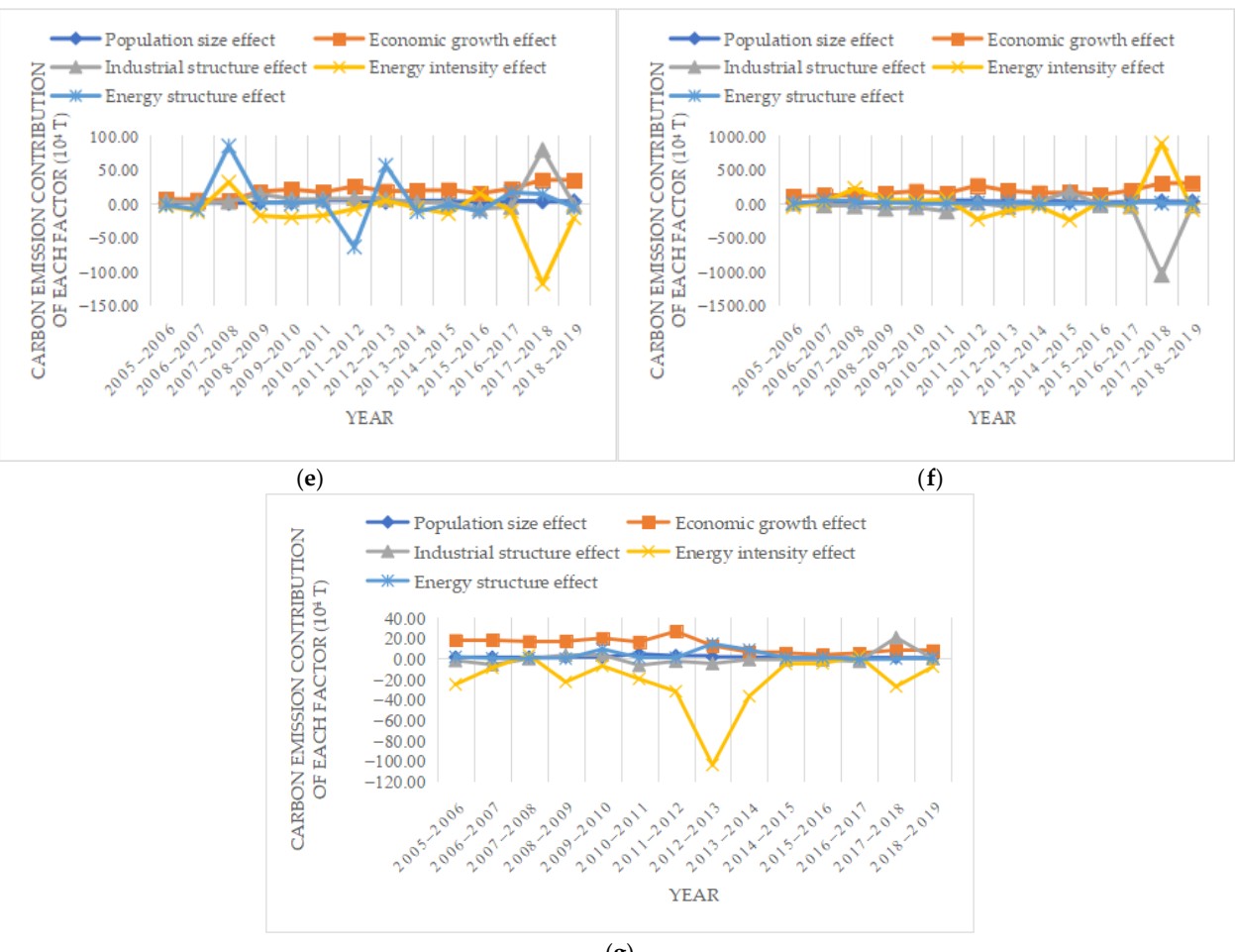

**Figure 3.** (**a**) FF; (**b**) MI; (**c**) MF; (**d**) PE; (**e**) CI; (**f**) TT; and (**g**) WS.

As can be seen from Figure 3 economic growth was the most important factor for the rapid increase in carbon emissions of each industry. The result is consistent with the previous studies for more than 30 provinces in China and 57 the Belt and Road countries [53,54]. It may be that economic growth promotes business activity through increased investment, purchases, consumption and energy consumption, which leads to increased pollution [55]. Population size had only a weak role in promoting carbon emissions. In addition, the world is facing population aging, which is no less harmful than environmental degradation. Therefore, the idea of controlling population growth to achieve consistent carbon emissions is not feasible. Since economic growth and population size have similar effects on carbon emissions of various industries, this study concentrated on an analysis of the effects of industry structure, energy intensity and energy structure on the carbon emissions of each industry.

4.2.1. Farming, Forestry, Animal Husbandry, Fishery and Water Conservancy Industry

The impact of industrial structure on agricultural carbon emissions was relatively small, and only significant in 2017–2018, indicating that change in the proportion of agricultural GDP in recent decades was not significant. One possible explanation is that Fujian is mountainous and close to the sea, and is rich in agricultural resources [56]. In the process of industrial upgrading, Fujian Province has retained its own advantages, so it generally transforms the secondary industry rather than agriculture into the tertiary industry. Energy intensity had a long-term inhibitory effect on carbon emissions, indicating that the energy utilization efficiency of agriculture was constantly improving. However, on the other hand, this was also because agricultural energy consumption was lower than that of industry

and construction. In agricultural carbon emissions, the role of the energy consumption structure was not obvious, which may be because the energy consumption structure of agriculture was very stable, mainly consuming coal and diesel. On the whole, agricultural carbon emissions showed a "U-shaped" curve, and the decline in carbon emissions was particularly obvious from 2012 to 2013. This was because during the "12th Five-Year Plan" period, Fujian Province was vigorously promoting energy-saving agricultural machinery, accelerating the elimination of old energy-consuming machinery, and generally promoting energy saving and emission reduction in agriculture [57]. Although some achievements have been made in reducing agricultural carbon emissions in Fujian Province, the rugged and hilly terrain of Fujian Province is not conducive to the popularization of large-scale modern agricultural appliances. This may become a difficult point for further agricultural modernization in Fujian Province.

### 4.2.2. Mining Industry

Industrial structure had a positive impact on carbon emissions before 2011. This may be due to the fact that Fujian Province was committed to planning the development of mineral resources throughout the province during the "11th Five-Year Plan" period, constantly and comprehensively improving the development and utilization of mineral resources, and increasing the scale of the mining industry [58]. The reason for expanding the scale of mining industry at this stage may be that the rapid economic development of Fujian Province needs a large amount of fossil energy as support. However, the effect of industrial structure was negative from 2011 to 2019, indicating that in this period Fujian Province was continuously reducing the scale of the mining industry and exploitation of local mineral resources. With the continuous strengthening of mineral resources management in Fujian Province, the scale of mining industry in Fujian Province may be further reduced in the future. The inhibition effect of industrial structure on carbon emissions may be further strengthened. Before 2013, the impact of energy intensity on carbon emissions from the mining industry was alternately positive and negative. From 2013 to 2019, most of the energy intensity effects were negative, which mainly inhibited the carbon emissions of the mining industry in Fujian Province. Energy structure had less influence in the research range, showing both positive and negative effects, indicating that the energy consumption structure of the mining industry had not been optimized, and that there is still much room for improvement in the future. In order to realize the decoupling of mining industry, Fujian Province needs to further improve the degree of mining intensification and high value. The mines with backward technology, poor safety production conditions and failing to meet the requirements of municipal green mines shall be eliminated and the mining license cancellation procedures shall be handled according to laws and regulations.

### 4.2.3. Manufacturing Industry

Industrial structure had a small impact on the carbon emissions of Fujian's manufacturing industry, having both inhibition and promotion effects on carbon emission. The industrial structure had a stable inhibitory effect on carbon emissions, after 2013. This may be related to the change of the secondary industry in Fujian Province. It is widely known that manufacturing industry accounts for the majority of the secondary industry. The proportion of secondary industry in Fujian Province showed a steady decline after 2014. Because during the "13th Five-Year Plan" period, Fujian was accelerating adjustment and optimization of the industrial structure, eliminating old-fashioned manufacturing methods, upgrading and transforming traditional manufacturing, and vigorously developing the tertiary industry [59]. Energy intensity had a boosting effect on carbon emissions in only a few years, and a more obvious suppressing effect on carbon emissions in most years, indicating that the utilization efficiency of energy by the manufacturing industry in Fujian Province had improved over the study period. Energy structure had both promoting and inhibiting effects on the manufacturing industry, mainly because the energy consumption of the manufacturing industry was relatively fixed and the energy structure adjustment is

insufficient [60]. Although many new energy sources have emerged in recent years, they have not yet been popularized. In general, the decoupling momentum of Fujian's manufacturing industry is good. To achieve further decarbonization of the manufacturing sector, the energy transition of the manufacturing sector needs to be addressed. On the one hand, the government can increase subsidies for manufacturing that uses green technologies, on the other hand, it can raise the entry threshold of industrial sectors to limit the development of high-carbon enterprises.

### 4.2.4. Electricity, Heat, Gas, Water Production and Supply Industries

Between 2005 and 2019, industrial structure had a large and basically inhibitory effect on carbon emissions. Because many provinces of China were continuously reducing the proportion of six high-energy-consuming industries, with the electricity, heat, gas and water production and supply industries bearing the brunt [61]. However, the economic development of Fujian Province has a broad prospect, and the demand for electric energy will remain at a high level in the future. Therefore, although the industrial structure is continuously optimized, the inhibition effect on carbon emissions is still limited. Energy intensity showed alternating inhibiting and promoting effects on the carbon emissions of this industry before 2011, and basically showed inhibiting effects after 2011. This was because the government of Fujian Province continuously strengthened energy-saving power generation dispatch and power demand-side management, and made major breakthroughs in energy construction during the "12th Five-Year Plan" and "13th Five- Year Plan" periods. Ningde and Fuqing nuclear power stations were successively built and put into operation [62], and the energy utilization rate of industries with high energy consumption was significantly improved. Energy structure had a significant impact on carbon emissions from the power, heat, gas and water production and supply industries, but the positive and negative effects did follow any obvious pattern, indicating that the effect of energy structure optimization in this industry is not obvious. One possible reason is that although the stage of rapid growth of thermal power in Fujian Province has passed and non-fossil energy power generation is in a stage of rapid development, thermal power generation is still in a dominant position in terms of the proportion of total power generation, and non-fossil energy power generation has still a large space for development. In view of this situation, Fujian Province should continue to strengthen the construction of various types of power sources, increase the proportion of clean energy in power generation, increase the investment in power grid construction, strengthen the research of energy storage equipment, and improve the efficiency of thermal power fuel conversion and utilization.

### 4.2.5. Construction Industry

Throughout the study period, economic growth promoted carbon emissions in the construction industry in Fujian Province, and this showed a fluctuating upward trend. Industrial structure essentially boosted carbon emissions before 2014, and suppressed them after 2014, with the exception of a large boosting effect on carbon emissions in 2017–2018, during which period the percentage of construction GDP in Fujian Province increased by 2.68%. Energy intensity had a greater inhibitory effect on carbon emissions during the study period, especially after 2017, indicating that the energy utilization rate of the construction industry in Fujian Province was constantly improving in recent years. But according to previous studies, the energy efficiency of the construction industry in China's provinces is still at a low level and needs to be improved further [63]. The energy structure effect curve fluctuated up and down around the *X*-axis, had obvious promoting effect on carbon emissions in 2007–2008 and 2012–2013, but obvious inhibiting effect in 2011–2012. The energy structure of the construction industry was relatively stable, mainly utilizing raw coal, gasoline and diesel. In general, Fujian Province has been promoting energy efficiency in the construction industry since the early 2000s, but the energy-saving effect was not obvious, and carbon emissions from the construction industry were still increasing year on year. Therefore, Fujian Province urgently needs to accelerate the low-carbon

transformation of the construction industry in order to effectively promote the realization of the goal of reaching the peak of carbon and carbon neutrality [64]. Fujian's construction industry can be reshaped in terms of product form, production mode, management mode, business mode and supervision mode and focused on new construction methods, green construction, clean energy, zero energy consumption buildings and other fields to carry out research on relevant technology routes, technology development, technology integration and other aspects.

### 4.2.6. Transportation, Storage, Post and Telecommunication Services

Economic growth was the main factor leading to the growth of carbon emissions in the transportation, storage, post and telecommunication services. This might be the result of the rapid development of the e-commerce industry in China in recent years [65]. Due to its superior geographical location and transportation advantages and good logistics foundation, Fujian Province is an important light textile industry production base and an international trade window province, and cross-border e-commerce has developed rapidly. Industrial structure had little impact on the carbon emissions of this industry, and had a significant inhibitory effect only in 2017–2018. This was due to the sharp drop in the proportion of Fujian's transportation, warehousing and postal industries from 5.58% to 3.56% in 2017–2018. Energy intensity promoted carbon emissions before 2011, and generally inhibited carbon emissions after 2011. However, it significantly promoted carbon emissions in 2017–2018, because the industrial structure significantly inhibited carbon emissions during this period. In contrast, energy intensity promoted carbon emissions to some extent. Energy structure had little impact on carbon emissions, because the energy structure of the transportation industry was fixed, mainly utilizing gasoline, kerosene, diesel and fuel oil. This is similar to the study of China by Yin et al. [66]. Their research shows that the transport sector is more difficult to decarbonise than other sectors of the economy, mainly because of its heavy reliance on petroleum products. Therefore, Fujian Province should attach importance to the research, development and production of new energy vehicles, and actively support the development of intelligent networked vehicles and hydrogen fuel cell vehicles in cities and regions with conditions, so as to reduce the carbon emissions of the transportation, storage, post and telecommunication services.

### 4.2.7. Wholesale, Retail Trade and Catering Services

Industrial structure and energy structure had just a small impact on the carbon emissions of wholesale, retail trade and catering services in Fujian Province during the study period. Energy intensity was the main factor inhibiting change in carbon emissions, especially in 2012–2013, indicating that the energy utilization rate of the industry was constantly improving. Energy structure mainly promoted carbon emissions, but the effect was not obvious, because the energy in this industry was still dominated by raw coal, gasoline, diesel and other fossil fuels. In recent years, Fujian Province continuously promoted low-carbon commercial pilot projects, selecting a number of commercial facilities such as catering institutions, hotels, shopping malls and tourist attractions, and guiding and encouraging them to improve their marketing concepts and models, strengthen the adoption of new technologies and products such as energy conservation and renewable energy, increase guidance of customer consumption behavior in order conserve resources and reduce carbon dioxide emissions [67]. Therefore, on the whole, the carbon emissions of this industry showed a declining trend. In general, the industry is currently in a decoupling state, and the impact of various factors on the industry's carbon emissions is small. In the future, the industry may be in a stable decoupling state for a long time.

### 4.3. Analysis of Industry Carbon Decoupling Efforts in Fujian Province

The LMDI index decomposition model can decompose carbon emissions to analyze the contribution of each influencing factor to carbon emissions, but it cannot objectively analyze the degree of the decoupling effort of each factor on carbon emissions. Therefore,

this study establishes a carbon emissions decoupling effort model based on the LMDI decomposition model in order to exclude the influence of economic growth on carbon emissions, and more objectively analyze the degree of decoupling effort of each influencing factor on carbon emissions in each industry. The results are shown in Figure 4.

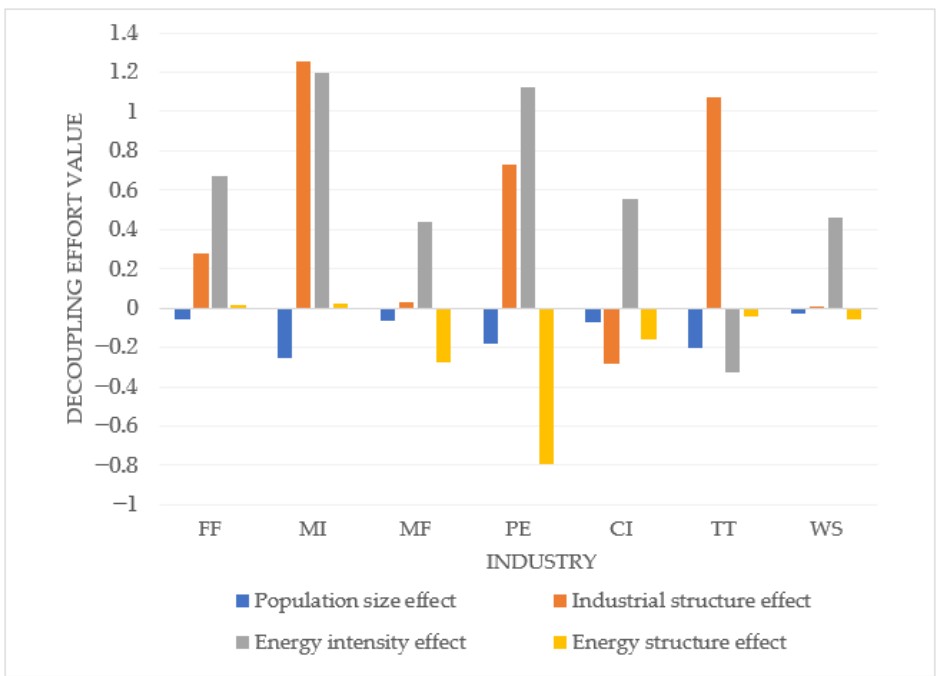

**Figure 4.** Decoupling effort index of various industries in Fujian Province 20052019.

It can be seen that from 2005 to 2019, the decoupling efforts of population size on all industries in Fujian Province were "no decoupling effort", because the number of permanent residents in Fujian Province was rising, and employees in all industries also showed an overall upward trend over the 15 years, therefore the population size placed a certain pressure on the decoupling of carbon emissions. Further, the empirical outcomes also support the past results [51,68]. However, inhibiting population growth will lead to population aging. Therefore, it is the key to advocate a simple, moderate, green and low-carbon lifestyle, actively promote the establishment of green and low-carbon demonstration as required, and improve the national awareness of energy conservation and low-carbon.

Industrial structure made decoupling efforts in most industries of Fujian. Moreover, the optimization of industrial structure is an effective measure to reduce carbon emissions, which has been proved by previous scholars, such as Zhang et al. [33]. In the mining industry, it showed "strong decoupling effort", and only in the construction industry, it showed "no decoupling effort", which indicated that Fujian Province made initial achievements in optimizing the industrial structure, and that the continued development of tertiary industry was conducive to the decoupling of economic growth and carbon emissions.

Energy intensity also showed positive decoupling effort in most industries. The result is consistent with the previous studies for Fujian provinces [68]. In particular, the industrial structure effect showed "strong decoupling effort" in the mining and electricity, heat, gas and water production and supply industries, but "no decoupling effort" in transportation, storage and postal services. This showed that Fujian Province invested in energy-saving and emission reduction technology research and development, and the energy utilization rates of most industries were effectively improved in the past 15 years, but the energy utilization rates of the transportation sector were still not high.

Because the carbon emission coefficients of different types of energy are different, increasing the use of clean energy such as renewable energy and natural gas can effectively reduce carbon emissions [69], so optimizing the energy structure is an effective emission

reduction measure. But, during the period 2005–2019, the decoupling effect of energy structure on industrial carbon emissions was insignificant, and its performance was not very stable. Even in manufacturing and other industries, the performance was "no decoupling effort", indicating that the effect of optimization of the energy structure of industries in Fujian Province was not obvious, that fossil fuels such as coal still dominated the energy consumption of most industries. Therefore, it is crucial to promote the research and development of renewable energy and energy storage technologies such as wind energy, solar energy, biomass energy, geothermal energy, marine energy and hydrogen energy, and accelerate the realization of new breakthroughs in nuclear energy and nuclear safety technology, smart grid and building energy conservation technology to reduce carbon emission in Fujian Province.

## 5. Conclusions and Policy Recommendations

### 5.1. Limitation and Future Studies

This paper still has some limitations, which makes future study possible. First, in this study, the factors affecting carbon emissions are divided into five factors: population size, economic growth, industrial structure, energy intensity and energy structure. However, these factors are relatively common in the study of carbon emission impact factors, and have not been expanded in combination with the specific situation of Fujian Province. In future research, we can consider the special geographical location of Fujian Province and introduce foreign trade and other factors to make the research more practical. Second, this paper only studies the carbon emissions of Fujian Province. Future research can analyze the impact of policies in different provinces on carbon emissions by comparing Fujian Province with other provinces to make the research results more convincing.

### 5.2. Conclusions

This study used the Tapio decoupling model to analyze the decoupling of seven key carbon emission industries in Fujian Province, and explored the factors driving carbon emissions in various industries in accordance with the LMDI decomposition model. Finally, based on the Tapio decoupling and LMDI decomposition models, a decoupling effort model for factors driving carbon emissions in various industries was established, and the following conclusions were drawn:

From the perspective of industry carbon emission characteristics, the total carbon emissions of industries in Fujian Province increased from 123.90 Mt to 278.11 Mt from 2005 to 2019, and were mainly concentrated on the manufacturing industry and production and supply of electric power, steam and hot water. Most of the industries gradually showed "strong decoupling" and "weak decoupling" from 2005 to 2019, indicating that the carbon emission reduction measures implemented in Fujian Province in the past showed initial results, but that the carbon emission decoupling status of some industries was not stable, and some industries even showed a tendency to change from "strong decoupling" to "weak decoupling". In general, Fujian Province should increase its carbon emission reduction efforts in the mining industry and the electricity, heat, gas and water production and supply industries, and it should not relax emission reduction initiatives for other industries.

From the contribution value of various factors driving industrial carbon emissions in Fujian Province, population scale and economic growth were the main drivers of increased emissions. In the long run, energy intensity effectively reduced carbon emissions in industries other than transportation, storage and postal services. Industrial structure had a greater inhibiting effect on carbon emissions in the mining industry and transportation, storage and postal industries, and a smaller inhibiting effect in other industries. Energy structure had a smaller effect on carbon emissions in industries in Fujian Province, because the energy consumption structure in Fujian Province was still dominated by fossil energy sources such as coal, while the use of renewable energy sources accounts for a relatively low share. From the decoupling effort value of various factors driving industrial carbon emissions, both energy structure and energy intensity made a greater decoupling effort on

carbon emissions. Industry structure made a smaller decoupling effort, and population size had not yet made a decoupling effort.

*5.3. Policy Recommendations*

Based on the above conclusions, the following suggestions are made on how to save energy and achieve emissions reduction in Fujian Province to promote sustainable development:

5.3.1. Dynamically Adjust the Emission Reduction Efforts of Different Industries

When formulating the industrial carbon emission reduction plan for Fujian Province, it is necessary to dynamically adjust the emission reduction efforts of different industries and focus on the manufacturing, mining and electricity, heat, gas and water production and supply industries that generate a large volume of carbon emissions and have not yet achieved a stable decoupling status. The government should also strengthen and improve relevant monitoring systems, impose severe punishments on individuals and enterprises that maliciously destroy the environment, monitor the emission indicators of individuals and enterprises in real time, immediately stop the production links with high pollution, investigate the unreasonable emission behaviors of pollutants, the production behaviors of products with low technology content and the behaviors of over-heavy industrialization to prevent some industries from rebounding after a period of "strong decoupling".

5.3.2. Strengthen Energy Technology Innovation and Improve Energy Utilization Efficiency

It is necessary to moderately slow the growth of GDP to reduce the intensity of carbon emissions and realize the transformation from a rough economy to a low-carbon economy. In the past, Fujian Province, like most eastern provinces, paid more attention to science and technology for promotion of economic development and acceleration of industrialization than to green technology. Therefore, when investing in science and technology, Fujian Province urgently needs to increase expenditure on green technologies to improve energy utilization efficiency and reduce energy intensity [70]. Specifically, it can be implemented from the two aspects of talents and enterprises. First of all, we should increase the intensity and scale of application-oriented personnel training, and better apply the scientific research achievements of the academic circle to practical production, so as to achieve a high degree of combination of industry, university and research. In addition, governments at all levels should increase support for relevant enterprises and encourage research and development units to actively develop technologies that can improve energy efficiency and the utilization of renewable and other clean energy sources.

5.3.3. Further Optimize the Industrial Structure

Compared with neighboring coastal provinces such as Guangdong and Zhejiang, which have better economic development, Fujian Province has a larger share of primary and secondary industries, but its share of tertiary industries is smaller. Therefore, Fujian Province needs to increase industrial restructuring and achieve the transformation from high-energy-consuming industries to high-tech industries. Specifically, it can be started from the following three aspects: (1) Differentiated carbon reduction actions should be carried out for different industries, and the high-energy-consuming industries, mainly power and steel, should be carried out collaborative governance of pollution reduction and carbon reduction. In the process of low-carbon industrial transformation, it is necessary to adapt to local conditions, vigorously develop local characteristic industries and leisure tourism services, and support the development of emerging new energy industries and high-tech industries. (2) We will implement the responsibility for carbon reduction in energy-intensive industries, and promptly eliminate outdated production capacity that does not meet energy consumption standards. (3) Using digital technology to promote the deep integration of green and environmental protection industry and information technology can promote the industry to replace low-end energy-intensive output with

high value. We will actively promote the construction of new energy industry innovation demonstration zones, further develop the development pattern of the new energy industry, and form new energy industry clusters.

### 5.3.4. Further Optimize the Energy Structure

Fujian Province should optimize the energy structure, gradually reduce the use of fossil energy, actively promote the construction of the Project of Natural Gas Transmission from West to East to use more environmentally friendly natural gas, and vigorously promote the development and utilization of wind, hydro, solar, geothermal, tidal and other renewable energy to curb carbon emissions [71], thus making full use of the advantages of its coastal areas, numerous mountains and rivers, and abundant water, heat and wind energy resources. In addition, Fujian Province is the core area of the construction of the 21st Century Maritime Silk Road, and has close cooperation with the "Belt and Road" countries [72]. This regional advantage can be fully utilized to promote cooperation between Fujian Province and the "Belt and Road" countries in the field of low-carbon science and technology, and accelerate the process of upgrading to green technology.

**Author Contributions:** Conceptualization, Q.J. and J.L.; methodology, J.L. and Q.W.; investigation, Q.J.; formal analysis, Q.J. and R.Z.; data curation, Q.J.; writing—original draft, Q.J. and J.L.; writing—review and editing, Q.W., H.F. and R.Z.; supervision, Q.J. and H.F. All authors have read and agreed to the published version of the manuscript.

**Funding:** This article was supported by National Social Science Foundation of China "The impact mechanism of organizational politics psychology on decision-making quality" (Grant No. 20BGL132).

**Institutional Review Board Statement:** Not applicable.

**Informed Consent Statement:** Not applicable.

**Data Availability Statement:** In this study, the data were mainly obtained from carbon emission accounts and datasets for emerging economies. Please check https://www.ceads.net/ (accessed on 5 March 2022).

**Acknowledgments:** The authors appreciate the anonymous reviewers for their constructive comments and suggestions that significantly improved the quality of this manuscript.

**Conflicts of Interest:** The authors declare no conflict of interest.

### Abbreviation

List of abbreviations in the study.

| Abbreviation | Full Name/Explanation |
| --- | --- |
| $CO_2$/C | Carbon dioxide |
| LMDI | Logarithmic mean Divisia index |
| IPCC | Intergovernmental Panel on Climate Change |
| EKC | Environmental Kuznets Curve |
| EU | European Union |
| ASEAN | Association of Southeast Asian Nations |
| ARDL | Auto-regressive distributive lag |
| SDA | Structural decomposition analysis |
| IDA | Index decomposition analysis |
| OLS | Ordinary Least Squares |
| GMM | Generalized method of moments |
| PECM | Panel error correction model |
| DSPM | Dynamic spatial panel model |
| POP/P | Population size |
| GDP | Gross domestic product |
| RQ | Regulatory quality |

| PS | Political stability |
|---|---|
| GE | Government effectiveness |
| EC/E | Energy consumption |
| URB | Urbanization rate |
| REC | Renewable energy consumption |
| TG | Technical progress |
| IS | Industrial structure |
| RD | Research and development investment |
| PE | Production efficiency |
| LF | Labor force |
| EG | Economic growth |
| EUE | Energy use efficiency |
| EI | Energy intensity |
| CEADs | China Emission Accounts and Datasets |
| GP | GDP per capita |
| ES | Energy structure |
| FF | Farming, forestry, animal husbandry, fishery and water conservancy |
| MI | Mining |
| MF | Manufacturing |
| PE | Production and supply of electric power, steam and hot water |
| CI | Construction |
| TT | Transportation, storage, post and telecommunication services |
| WS | Wholesale, retail trade and catering services |
| OECD | Organization for Economic Cooperation and Development |

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
