# Peer review of "Demystifying the Economic Growth and CO2 Nexus in Fujian’s Key Industries Based on Decoupling and LMDI Model"

_sustainability, doi:10.3390/su15043863_

Round 1

Reviewer 1 Report

This is an interesting paper focusing on economic growth and CO2 nexus in Fujian's key industries. I have the following comments/suggestions with the sole aim of improving the final version of this study. 

In the introduction, while trying to justify this study, you may need to add why Tapio decoupling model and LMDI index decomposition methods were used instead of the traditional econometric model..

Kindly add the map of Fujian, China if available. 

In the methods, you are expected to describe data used in this study (cross-sectional or secondary data?). Get your readers familiar with your study data sources with detail description.

GIve the meaning of CEAD and other acronyms at first mention in this manuscript. 

Apart from the figures describing the results of this study, it expedient that authors present a robust detail of the results in which the figures will support and not the other way.

In the discussion section, I am surprised that no empirical studies were used to discuss the results in this study. You expected to rigorously discuss the findings in this research using other empirical studies in literature to either support or refute your findings. Please, check this. 

Before the conclusion section, kindly present a sub-section on "limitations of the study" and "areas for further research"

Thank you.

Author Response

Response to Reviewer 1 Comments

Point 1: In the introduction, while trying to justify this study, you may need to add why Tapio decoupling model and LMDI index decomposition methods were used instead of the traditional econometric model.

Response 1: Based on the literature that used decoupling method in the past, we summarized the advantages of decoupling method compared with traditional measurement methods (line 89): Compared with traditional measurement methods, Tapio decoupling and LMDI method is more concise in calculation, and can dynamically and real-time reflect the annual contribution of various influencing factors to the change of carbon emissions, making the results more intuitive.

Point 2: Kindly add the map of Fujian, China if available.

Response 2: The geographical location map of Fujian Province is attached (figure 1).

Point 3: In the methods, you are expected to describe data used in this study (cross-sectional or secondary data?). Get your readers familiar with your study data sources with detail description.

Response 3: We have detailed the data as follows (Sections 3.2): The years 2005-2019 were selected as the study period, and the industry energy consumption data and carbon emissions data were secondary data mainly obtained from the China Emission Accounts and Datasets (CEADs). Besides, the website link has been attached at the end of the article.

Point 4: Give the meaning of CEAD and other acronyms at first mention in this manuscript.

Response 4: The full name of the acronym has been attached when it first appeared. And the appendix summarizes all abbreviations.

Point 5: Apart from the figures describing the results of this study, it expedient that authors present a robust detail of the results in which the figures will support and not the other way.

Response 5: The data and conclusions of this study are supported by the past policies and plans of Fujian Province. This has been presented in the paper.

Point 6: In the discussion section, I am surprised that no empirical studies were used to discuss the results in this study. You expected to rigorously discuss the findings in this research using other empirical studies in literature to either support or refute your findings. Please, check this.

Response 6: We have added more than ten articles in the results section to strictly discuss the findings of this study to support our view. Literature review are improved by adding more recent research as following:

[47] Guan, D.; Meng, J.; Reiner, D.M.; Zhang, N.; Shan, Y.; Mi, Z.; Shao, S.; Liu, Z.; Zhang, Q.; Davis, S.J. Structural decline in China’s CO2 emissions through transitions in industry and energy systems. Nat. Geosci. 2018, 8, 551-555.

[48] Li, K.; Lin, B. Economic growth model, structural transformation, and green productivity in China. Appl. Energy. 2017, 187, 489-500.

[49] Han, H.; Zhong, Z.; Guo, Y.; Xi, F.; Liu, S. Coupling and decoupling effects of agricultural carbon emissions in China and their driving factors. Environ. Sci. Pollut. Res. 2018, 25, 25280-25293.

[50] Dong, B.; Ma, X.; Zhang, Z.; Zhang, H.; Chen, R.; Song, Y.; Shen, M.; Xiang, R. Carbon emissions, the industrial structure and economic growth: Evidence from heterogeneous industries in China. Environmental Pollution. 2020, 262, 114322.

[51]Yu, J.; Wang, Y.; Yu, F.; Luo, J.; Lai, W. Decoupling between resources and environment and economic growth in Fujian Province, China from the perspective of" water-energy-carbon" consumption. J. Appl. Ecol. 2021, 11, 3845-3855. 

[53] Zhao, X.; Burnett, J.W.; Fletcher, J.J. Spatial analysis of China province-level CO2 emission intensity. Renew. Sust. Energ. Rev. 2014, 33, 1-10.

[54] Jiang, Q.; Rahman, Z.U.; Zhang, X.; Guo, Z.; Xie, Q. An assessment of the impact of natural resources, energy, institutional quality, and financial development on CO2 emissions: Evidence from the B&R nations. Resour. Policy. 2022, 76, 102716.

[55] Baloch, M.A.; Zhang, J.; Iqbal, K.; Iqbal, Z. The effect of financial development on ecological footprint in BRI countries: evidence from panel data estimation. Environ. Sci. Pollut. Res. 2019, 6, 6199-6208.

[58] Fujian Provincial People’s Government. General Office of the Fujian Provincial People's Government on the issuance of the "13th Five-Year Plan" for the development of modern service industry special planning notice. Available online: https://www.fujian.gov.cn/zwgk/ghjh/ghxx/ 201606/t20160606_1200706.htm (accessed on 6 June 2016).

[60] Li, S. Research on Energy Efficiency Evaluation of Chinese construction industry from Economic and environmental perspective; China University of Petroleum: Qingdao, China, 2021.

[61] Yin, X.; Chen, W.; Eom, J.; Clarke, L.E.; Kim, S.H.; Patel, P.L.; Yu, S.; Kyle, G.P. China's transportation energy consumption and CO2 emissions from a global perspective. Energy Policy. 2015, 82, 233-248.

[63] Zhong, Y. Study on the Influencing Factors and Decoupling Characteristics of Carbon Emission in Fujian Province; Fujian Normal University: Fujian, China, 2018.

In addition, we also enrich the policy recommendations.

Point 7: Before the conclusion section, kindly present a sub-section on "limitations of the study" and "areas for further research".

Response 7: The limitations of the study have been added at the end of the paper (Section 6): This paper still has some limitations, which makes future study possible. First, in this study, the factors affecting carbon emissions are divided into five factors: population size, economic growth, industrial structure, energy intensity and energy structure. However, these factors are relatively common in the study of carbon emission impact factors, and have not been expanded in combination with the specific situation of Fujian Province. In future research, we can consider the special geographical location of Fujian Province and introduce foreign trade and other factors to make the research more practical. Second, this paper only studies the carbon emissions of Fujian Province. Future research can analyze the impact of policies in different provinces on carbon emissions by comparing Fujian Province with other provinces to make the research results more convincing.

Thanks for your time and patience again!

Reviewer 2 Report

Tapio Decoupling Model proposed in 2005 using the change in growth rate to analyze the decoupling relationship between carbon emissions and economic growth, which is a safe and traditional methodology. The study has to provide more creative view point, valuable policy recommendations to apparent academic contribution. While this version is short of it.

For example, manufacturing industry structure is complex and change with time. Now this discussion is simple and some causal relationship is open to discussion, such as “The inhibiting effect in 2013–2017 was relatively obvious, indicating that Fujian was accelerating adjustment and optimization of the industrial structure, eliminating old-fashioned manufacturing methods, upgrading and transforming traditional manufacturing, and vigorously developing the tertiary industry”. Above statement need more evidence to support the author’s view. The other industries have the same matters.

Because of above matter, the analysis of Industry carbon decoupling efforts in Fujian Province is not creative thinking.

       Tapio Decoupling Model proposed in 2005 using the change in growth rate to analyze the decoupling relationship between carbon emissions and economic growth, which is a safe and traditional methodology. The study has to provide more creative view point, valuable policy recommendations to apparent academic contribution. While this version is short of it.

For example, manufacturing industry structure is complex and change with time. Now this discussion is simple and some causal relationship is open to discussion, such as “The inhibiting effect in 2013–2017 was relatively obvious, indicating that Fujian was accelerating adjustment and optimization of the industrial structure, eliminating old-fashioned manufacturing methods, upgrading and transforming traditional manufacturing, and vigorously developing the tertiary industry”. Above statement need more evidence to support the author’s view. The other industries have the same matters.

Because of above matter, the analysis of Industry carbon decoupling efforts in Fujian Province is not creative thinking.

Author Response

Response to Reviewer 2 Comments

Point 1: The study has to provide more creative view point, valuable policy recommendations to apparent academic contribution. While this version is short of it.

Response 1: Thank you for your valuable suggestions, which are very useful for the improvement of paper. We have enriched our policy recommendations to make them more detailed and targeted.

For example, in response to the unstable state of industrial decoupling in Fujian Province, we proposed to dynamically adjust the emission reduction efforts of different industries and strengthen government supervision. We have also enriched the means of government supervision. The government should also strengthen and improve the relevant monitoring system, severely punish individuals and enterprises that maliciously damage the environment, monitor individual and enterprise emission indicators in real time, immediately stop highly polluting production links, and investigate unreasonable discharge of pollutants.

For the proposal of reducing the intensity of carbon emissions, we also elaborated in detail from individuals and companies. We also elaborated on the suggestion of adjusting the industrial structure from three aspects.

For the suggestion of adjusting the energy structure, we put forward some specific suggestions according to the location advantage of Fujian Province. For example, Fujian Province's sufficient geothermal energy and wind energy are used to optimize the energy structure. And take advantage of Fujian Province as the core region of the the Belt and Road to carry out new energy cooperation with the Belt and Road countries. Please refer to the fifth part of this article for details.

Point 2: The manufacturing industry structure is complex and change with time. Now this discussion is simple and some causal relationship is open to discussion, such as “The inhibiting effect in 2013-2017 was relatively obvious, indicating that Fujian was accelerating adjustment and optimization of the industrial structure, eliminating old-fashioned manufacturing methods, upgrading and transforming traditional manufacturing, and vigorously developing the tertiary industry”. Above statement need more evidence to support the author’s view. The other industries have the same matters.

Response 2: Thank you for pointing out an important question. We have added more than ten articles in the results section to strictly discuss the findings of this study to support our view. Literature review are improved by adding more recent research as following:

[47] Guan, D.; Meng, J.; Reiner, D.M.; Zhang, N.; Shan, Y.; Mi, Z.; Shao, S.; Liu, Z.; Zhang, Q.; Davis, S.J. Structural decline in China’s CO2 emissions through transitions in industry and energy systems. Nat. Geosci. 2018, 8, 551-555.

[48] Li, K.; Lin, B. Economic growth model, structural transformation, and green productivity in China. Appl. Energy. 2017, 187, 489-500.

[49] Han, H.; Zhong, Z.; Guo, Y.; Xi, F.; Liu, S. Coupling and decoupling effects of agricultural carbon emissions in China and their driving factors. Environ. Sci. Pollut. Res. 2018, 25, 25280-25293.

[50] Dong, B.; Ma, X.; Zhang, Z.; Zhang, H.; Chen, R.; Song, Y.; Shen, M.; Xiang, R. Carbon emissions, the industrial structure and economic growth: Evidence from heterogeneous industries in China. Environmental Pollution. 2020, 262, 114322.

[51]Yu, J.; Wang, Y.; Yu, F.; Luo, J.; Lai, W. Decoupling between resources and environment and economic growth in Fujian Province, China from the perspective of" water-energy-carbon" consumption. J. Appl. Ecol. 2021, 11, 3845-3855.

[53] Zhao, X.; Burnett, J.W.; Fletcher, J.J. Spatial analysis of China province-level CO2 emission intensity. Renew. Sust. Energ. Rev. 2014, 33, 1-10.

[54] Jiang, Q.; Rahman, Z.U.; Zhang, X.; Guo, Z.; Xie, Q. An assessment of the impact of natural resources, energy, institutional quality, and financial development on CO2 emissions: Evidence from the B&R nations. Resour. Policy. 2022, 76, 102716.

[55] Baloch, M.A.; Zhang, J.; Iqbal, K.; Iqbal, Z. The effect of financial development on ecological footprint in BRI countries: evidence from panel data estimation. Environ. Sci. Pollut. Res. 2019, 6, 6199-6208.

[58] Fujian Provincial People’s Government. General Office of the Fujian Provincial People's Government on the issuance of the "13th Five-Year Plan" for the development of modern service industry special planning notice. Available online: https://www.fujian.gov.cn/zwgk/ghjh/ghxx/ 201606/t20160606_1200706.htm (accessed on 6 June 2016).

[60] Li, S. Research on Energy Efficiency Evaluation of Chinese construction industry from Economic and environmental perspective; China University of Petroleum: Qingdao, China, 2021.

[61] Yin, X.; Chen, W.; Eom, J.; Clarke, L.E.; Kim, S.H.; Patel, P.L.; Yu, S.; Kyle, G.P. China's transportation energy consumption and CO2 emissions from a global perspective. Energy Policy. 2015, 82, 233-248.

[63] Zhong, Y. Study on the Influencing Factors and Decoupling Characteristics of Carbon Emission in Fujian Province; Fujian Normal University: Fujian, China, 2018.

The ideas in the paper, such as “The inhibiting effect in 2013-2017 was relatively obvious, indicating that Fujian was accelerating adjustment and optimization of the industrial structure, eliminating old-fashioned manufacturing methods, upgrading and transforming traditional manufacturing, and vigorously developing the tertiary industry”, is derived from model results and is consistent with the “13th Five-Year” Plan of Fujian Province. We add literature or policies that support the view.

In the analysis results of other industries, we also try to cite policy and empirical literature to support and discuss these views. For details, please refer to the documents added in the article. We have marked these changes.

Thanks for your time and patience again!

Round 2

Reviewer 1 Report

Thank you for revising this manuscript. I feel you just need to structure this paper properly for clarity.

From the discussion of results, you need to make the section clear and visible as "results and discussion" and not "results and analysis". Kindly adjust this.

The limitations of the study should come before conclusion section and not after. Please, do that.

Thank you.

Author Response

Point 1: Thank you for revising this manuscript. I feel you just need to structure this paper properly for clarity. From the discussion of results, you need to make the section clear and visible as "results and discussion" and not "results and analysis". Kindly adjust this.

Response 1: Thank you for your valuable suggestions, which are very useful for the improvement of paper. We have shown the fourth part as “results and discussion” rather than “results and analysis”. In addition, we have substantially increased our viewpoint and discussions based on empirical results. It includes the reasons for the formation of the results, the possible trend of change in the future, and the policy priorities and difficulties of various industry decoupling. The major changes are in Section 4 (Results and Discussion), particularly Section 4.2. The details are as follows:

  1. The first is about the modification of 4.1 (Current Situation of Carbon Emissions and Decoupling of Industries in Fujian Province).

  • Our results are similar to those of Guan et al. [47] and Li et al. [48] regarding the trend of carbon emission change in Fujian Province. We further discussed the relationship between the change trend of carbon emissions in Fujian Province and the EKC curve, and discussed the causes of the curve change. The main additions are as follows: (line 344): In the early stage of industrialization, most regions and countries used basic energy in large quantities in order to Vigorously develop their economy With the economic development reaching a certain level, regions and countries will pay more attention to environmental protection, improve energy efficiency and carry out industrial trans-formation. The trend of carbon emissions in Fujian Province is also consistent with the EKC hypothesis.

  • For the decoupling of agricultural carbon emissions in Fujian Province, our conclusion is consistent with Han et al. [49]. We also discussed several key points for reducing agricultural pollution. The main additions are as follows: (line 363): Although the industry is still in the decoupling state, attention needs to be paid to improving quality and efficiency. In the future, Fujian Province should not only pay attention to carbon emissions caused by agricultural energy use, but also reduce environmental pollution caused by the use of chemical fertilizers, pesticides and improper treatment of crops such as straw.

  • For the decoupling of manufacturing carbon emissions in Fujian Province, our conclusion is similar to Yu et al. [49]. We added the following dialogue (line 388): Although the decarbonization of Fujian's manufacturing industry is good, the industry is one of the main industries of carbon emissions, and it is still challenging to maintain the decoupling status.
  • For the decoupling of the electricity, heat, gas and water production and supply industries, based on the comparison with other literature results, we further put forward our views and dialogue on the carbon emission reduction of the industry(line 400): In addition, as an important energy infrastructure industry, electric power is both the energy supply side and the largest energy consumption field. To achieve the goal of carbon neutralization, low carbonization of power supply structure is the key path. So Fujian Province should pay particular attention to the issue of carbon emissions in this industry.

  1. The second is about the modification of 4.2 (Analysis of Influencing Factors of Industrial Carbon Emissions in Fujian Province).

  • Population size had only a weak role in promoting carbon emissions. We further increase the discussion (line 424): The world is facing population aging, which is no less harmful than environmental degradation. Therefore, the idea of controlling population growth to achieve consistent carbon emissions is not feasible.

  • For the analysis of the decomposition results of the factors affecting agricultural carbon emissions, we have added discussions, views and dialogues.

First, the reason why the change of industrial structure has little impact on carbon emissions (line 442): One possible explanation is that Fujian is mountainous and close to the sea, and is rich in agricultural resources [56]. In the process of industrial upgrading, Fujian Province has retained its own advantages, so it generally transforms the secondary industry rather than agriculture into the tertiary industry.

In addition, the difficulties of further agricultural decarbonization in Fujian Province are also discussed (line 456): Although some achievements have been made in reducing agricultural carbon emissions in Fujian Province, the rugged and hilly terrain of Fujian Province is not conducive to the popularization of large-scale modern agricultural appliances. This may become a difficult point for further agricultural modernization in Fujian Province.

  • For the analysis of the decomposition results of the factors affecting mining industrial carbon emissions, we have added discussions, views and dialogues. 

The reasons for expanding the scale of mining industry in Fujian Province during the "Eleventh Five-Year Plan" are discussed (line 466): The reason for expanding the scale of mining industry at this stage may be that the rapid economic development of Fujian Province needs a large amount of fossil energy as support.

Added a dialogue on the future changes in the industrial structure of the mining industry in Fujian Province (line 471): With the continuous strengthening of mineral resources management in Fujian Province, the scale of mining industry in Fujian Province may be further reduced in the future. The inhibition effect of industrial structure on carbon emissions may be further strengthened.

We also put forward our own views on the decoupling of carbon emissions in the mining industry (line 480): In order to realize the decoupling of mining industry, Fujian Province needs to further improve the degree of mining intensification and high value. The mines with back-ward technology, poor safety production conditions and failing to meet the requirements of municipal green mines shall be eliminated and the mining license cancellation procedures shall be handled according to laws and regulations.

  • For the analysis of the decomposition results of the factors affecting manufacturing industrial carbon emissions, we have added discussions, views and dialogues.

We discussed the reasons why the manufacturing industry structure inhibits carbon emissions (line 489): This may be related to the change of the secondary industry in Fujian Province. It is widely known that manufacturing industry accounts for the majority of the secondary industry. The proportion of secondary industry in Fujian Province showed a steady decline after 2014 .

We discussed the reasons why the manufacturing energy structure inhibits carbon emissions (line 60): Energy structure had both promoting and inhibiting effects on the manufacturing industry, mainly because the energy consumption of the manufacturing industry was relatively fixed and the energy structure adjustment is insufficient [60].

We also put forward policy views on further decarbonization of manufacturing industry (line 504): In general, the decoupling momentum of Fujian's manufacturing industry is good. To achieve further decarbonization of the manufacturing sector, the energy transition of the manufacturing sector needs to be addressed. On the one hand, the government can increase subsidies for manufacturing that uses green technologies, on the other hand, it can raise the entry threshold of industrial sectors to limit the development of high-carbon enterprises.

  • For the analysis of the decomposition results of the factors affecting electricity, heat, gas and water production and supply industrial carbon emissions, we have added discussions, views and dialogues.

The industrial structure has a great impact on carbon emissions and is basically a cause of inhibition (line 514): Because Fujian many provinces of China Province was continuously reducing the proportion of six high-energy-consuming industries, with the electricity, heat, gas and water production and supply industries bearing the brunt [61]. However, the economic development of Fujian Province has a broad prospect, and the demand for electric energy will remain at a high level in the future. Therefore, although the industrial structure is continuously optimized, the inhibition effect on carbon emissions is still limited.

We discussed the possible reasons why the energy structure did not contribute to the decarbonization of the industry, and put forward our own views (line 531): One possible reason is that although the stage of rapid growth of thermal power in Fujian Province has passed and non-fossil energy power generation is in a stage of rapid development, thermal power generation is still in a dominant position in terms of the proportion of total power generation, and non-fossil energy power generation has still a large space for development. In view of this situation, Fujian Province should continue to strengthen the construction of various types of power sources, in-crease the proportion of clean energy in power generation, increase the investment in power grid construction, strengthen the research of energy storage equipment, and improve the efficiency of thermal power fuel conversion and utilization.

  • For the analysis of the decomposition results of the factors affecting construction industrial carbon emissions, we have added discussions, views and dialogues.

The importance of decoupling carbon emissions in the construction industry has been verified by reference. The details are as follows (line 558): Therefore, Fujian Province urgently needs to accelerate the low-carbon transformation of the construction industry in order to effectively promote the realization of the goal of reaching the peak of carbon and carbon neutrality [64].

We increased the dialogue on emission reduction in the construction industry (line 560): Fujian's construction industry can be reshaped in terms of product form, production mode, management mode, business mode and supervision mode and focused on new construction methods, green construction, clean energy, zero energy consumption buildings and other fields to carry out research on relevant technology routes, technology development, technology integration and other aspects.

  • For the analysis of the decomposition results of the factors affecting transportation, storage, post and telecommunication services industrial carbon emissions, we have added discussions, views and dialogues.

Reasons for economic growth to promote carbon emissions of the industry (line 568): This might be the result of the rapid development of the e-commerce industry in China in recent years [65].

We added the discussion on the development of e-commerce in Fujian Province (line 570): Due to its superior geographical location and transportation advantages and good logistics foundation, Fujian Province is an important light textile industry production base and an international trade window province, and cross-border e-commerce has developed rapidly.

We increased the dialogue on emission reduction in transportation industry (line 560): Therefore, Fujian Province should attach importance to the research, development and production of new energy vehicles, and actively support the development of intelligent networked vehicles and hydrogen fuel cell vehicles in cities and regions with conditions, so as to reduce the carbon emissions of the transportation, storage, post and telecommunication services.

  • We have increased our view on the future decoupling of carbon emissions in the construction industry(line 605): In general, the industry is currently in a decoupling state, and the impact of various factors on the industry’s carbon emissions is small. In the future, the industry may be in a stable decoupling state for a long time.

  1. The third is about the modification of 4.3 (Analysis of Industry Carbon Decoupling Efforts in Fujian Province).

  • The first is the viewpoint that population growth will promote carbon emissions (line 621): However, inhibiting population growth will lead to population aging. Therefore, it is the key to advocate a simple, moderate, green and low-carbon lifestyle, actively pro-mote the establishment of green and low-carbon demonstration as required, and improve the national awareness of energy conservation and low-carbon.

  • We cite empirical literature to support our research results about industrial structure. The details are as follows (line 626): Moreover, the optimization of industrial structure is an effective measure to reduce carbon emissions, which has been proved by previous scholars, such as Zhang et al. [33].

  • We cite empirical literature to support our research results about energy intensity. The details are as follows (line 634): Energy intensity also showed positive decoupling effort in most industries, . The result is consistent with the previous studies for Fujian provinces [68].

  • We discussed the reasons why the energy structure affects carbon emissions(line 645): Because the carbon emission coefficients of different types of energy are different, increasing the use of clean energy such as renewable energy and natural gas can effectively reduce carbon emissions [69], so optimizing the energy structure is an effective emission reduction measure.

  • We added a dialogue on energy structure optimization(line 653): Therefore, it is crucial to promote the research and development of renewable energy and energy storage technologies such as wind energy, solar energy, biomass energy, geothermal energy, marine energy and hydrogen energy, and accelerate the realization of new breakthroughs in nuclear energy and nuclear safety technology, smart grid and building energy conservation technology to reduce carbon emission in Fujian Province.

Point 2: The limitations of the study should come before conclusion section and not after. Please, do that.

Response 2: Thank you for your suggestion. We have put the limitations of this study before the conclusion.

We have marked all changes in the manuscript. Thanks for your time and patience again!

Reviewer 2 Report

The revise indeed increased many literature, but their discussion and dialogue were  meager. It lacks own point of view, which has been remind in the first review. While it did no collct in this revise.

The authors have improve their discussion and dialogueby a wide margin in the next revision. 

Author Response

Point: The revise indeed increased many literature, but their discussion and dialogue were  meager. It lacks own point of view, which has been remind in the first review. While it did no collct in this revise.

The authors have improve their discussion and dialogue by a wide margin in the next revision.

Response: Thank you for your valuable suggestions, which are very useful for the improvement of paper. On the basis of adding more than ten references for the first time, we have added some documents to support our research results. As suggested, we have substantially increased our viewpoint, discussions and dialogue based on empirical results. It includes the reasons for the formation of the results, the possible trend of change in the future, and the policy priorities and difficulties of various industry decoupling. The major changes are in Section 4 (Results and Discussion), particularly Section 4.2. The details are as follows:

  1. The first is about the modification of 4.1 (Current Situation of Carbon Emissions and Decoupling of Industries in Fujian Province).

  • Our results are similar to those of Guan et al. [47] and Li et al. [48] regarding the trend of carbon emission change in Fujian Province. We further discussed the relationship between the change trend of carbon emissions in Fujian Province and the EKC curve, and discussed the causes of the curve change. The main additions are as follows: (line 344): In the early stage of industrialization, most regions and countries used basic energy in large quantities in order to Vigorously develop their economy With the economic development reaching a certain level, regions and countries will pay more attention to environmental protection, improve energy efficiency and carry out industrial trans-formation. The trend of carbon emissions in Fujian Province is also consistent with the EKC hypothesis.

  • For the decoupling of agricultural carbon emissions in Fujian Province, our conclusion is consistent with Han et al. [49]. We also discussed several key points for reducing agricultural pollution. The main additions are as follows: (line 363): Although the industry is still in the decoupling state, attention needs to be paid to improving quality and efficiency. In the future, Fujian Province should not only pay attention to carbon emissions caused by agricultural energy use, but also reduce environmental pollution caused by the use of chemical fertilizers, pesticides and improper treatment of crops such as straw.

  • For the decoupling of manufacturing carbon emissions in Fujian Province, our conclusion is similar to Yu et al. [49]. We added the following dialogue (line 388): Although the decarbonization of Fujian's manufacturing industry is good, the industry is one of the main industries of carbon emissions, and it is still challenging to maintain the decoupling status.
  • For the decoupling of the electricity, heat, gas and water production and supply industries, based on the comparison with other literature results, we further put forward our views and dialogue on the carbon emission reduction of the industry(line 400): In addition, as an important energy infrastructure industry, electric power is both the energy supply side and the largest energy consumption field. To achieve the goal of carbon neutralization, low carbonization of power supply structure is the key path. So Fujian Province should pay particular attention to the issue of carbon emissions in this industry.

  1. The second is about the modification of 4.2 (Analysis of Influencing Factors of Industrial Carbon Emissions in Fujian Province).

  • Population size had only a weak role in promoting carbon emissions. We further increase the discussion (line 424): The world is facing population aging, which is no less harmful than environmental degradation. Therefore, the idea of controlling population growth to achieve consistent carbon emissions is not feasible.

  • For the analysis of the decomposition results of the factors affecting agricultural carbon emissions, we have added discussions, views and dialogues.

First, the reason why the change of industrial structure has little impact on carbon emissions (line 442): One possible explanation is that Fujian is mountainous and close to the sea, and is rich in agricultural resources [56]. In the process of industrial upgrading, Fujian Province has retained its own advantages, so it generally transforms the secondary industry rather than agriculture into the tertiary industry.

In addition, the difficulties of further agricultural decarbonization in Fujian Province are also discussed (line 456): Although some achievements have been made in reducing agricultural carbon emissions in Fujian Province, the rugged and hilly terrain of Fujian Province is not conducive to the popularization of large-scale modern agricultural appliances. This may become a difficult point for further agricultural modernization in Fujian Province.

  • For the analysis of the decomposition results of the factors affecting mining industrial carbon emissions, we have added discussions, views and dialogues. 

The reasons for expanding the scale of mining industry in Fujian Province during the "Eleventh Five-Year Plan" are discussed (line 466): The reason for expanding the scale of mining industry at this stage may be that the rapid economic development of Fujian Province needs a large amount of fossil energy as support.

Added a dialogue on the future changes in the industrial structure of the mining industry in Fujian Province (line 471): With the continuous strengthening of mineral resources management in Fujian Province, the scale of mining industry in Fujian Province may be further reduced in the future. The inhibition effect of industrial structure on carbon emissions may be further strengthened.

We also put forward our own views on the decoupling of carbon emissions in the mining industry (line 480): In order to realize the decoupling of mining industry, Fujian Province needs to further improve the degree of mining intensification and high value. The mines with back-ward technology, poor safety production conditions and failing to meet the requirements of municipal green mines shall be eliminated and the mining license cancellation procedures shall be handled according to laws and regulations.

  • For the analysis of the decomposition results of the factors affecting manufacturing industrial carbon emissions, we have added discussions, views and dialogues.

We discussed the reasons why the manufacturing industry structure inhibits carbon emissions (line 489): This may be related to the change of the secondary industry in Fujian Province. It is widely known that manufacturing industry accounts for the majority of the secondary industry. The proportion of secondary industry in Fujian Province showed a steady decline after 2014 .

We discussed the reasons why the manufacturing energy structure inhibits carbon emissions (line 60): Energy structure had both promoting and inhibiting effects on the manufacturing industry, mainly because the energy consumption of the manufacturing industry was relatively fixed and the energy structure adjustment is insufficient [60].

We also put forward policy views on further decarbonization of manufacturing industry (line 504): In general, the decoupling momentum of Fujian's manufacturing industry is good. To achieve further decarbonization of the manufacturing sector, the energy transition of the manufacturing sector needs to be addressed. On the one hand, the government can increase subsidies for manufacturing that uses green technologies, on the other hand, it can raise the entry threshold of industrial sectors to limit the development of high-carbon enterprises.

  • For the analysis of the decomposition results of the factors affecting electricity, heat, gas and water production and supply industrial carbon emissions, we have added discussions, views and dialogues.

The industrial structure has a great impact on carbon emissions and is basically a cause of inhibition (line 514): Because Fujian many provinces of China Province was continuously reducing the proportion of six high-energy-consuming industries, with the electricity, heat, gas and water production and supply industries bearing the brunt [61]. However, the economic development of Fujian Province has a broad prospect, and the demand for electric energy will remain at a high level in the future. Therefore, although the industrial structure is continuously optimized, the inhibition effect on carbon emissions is still limited.

We discussed the possible reasons why the energy structure did not contribute to the decarbonization of the industry, and put forward our own views (line 531): One possible reason is that although the stage of rapid growth of thermal power in Fujian Province has passed and non-fossil energy power generation is in a stage of rapid development, thermal power generation is still in a dominant position in terms of the proportion of total power generation, and non-fossil energy power generation has still a large space for development. In view of this situation, Fujian Province should continue to strengthen the construction of various types of power sources, in-crease the proportion of clean energy in power generation, increase the investment in power grid construction, strengthen the research of energy storage equipment, and improve the efficiency of thermal power fuel conversion and utilization.

  • For the analysis of the decomposition results of the factors affecting construction industrial carbon emissions, we have added discussions, views and dialogues.

The importance of decoupling carbon emissions in the construction industry has been verified by reference. The details are as follows (line 558): Therefore, Fujian Province urgently needs to accelerate the low-carbon transformation of the construction industry in order to effectively promote the realization of the goal of reaching the peak of carbon and carbon neutrality [64].

We increased the dialogue on emission reduction in the construction industry (line 560): Fujian's construction industry can be reshaped in terms of product form, production mode, management mode, business mode and supervision mode and focused on new construction methods, green construction, clean energy, zero energy consumption buildings and other fields to carry out research on relevant technology routes, technology development, technology integration and other aspects.

  • For the analysis of the decomposition results of the factors affecting transportation, storage, post and telecommunication services industrial carbon emissions, we have added discussions, views and dialogues.

Reasons for economic growth to promote carbon emissions of the industry (line 568): This might be the result of the rapid development of the e-commerce industry in China in recent years [65].

We added the discussion on the development of e-commerce in Fujian Province (line 570): Due to its superior geographical location and transportation advantages and good logistics foundation, Fujian Province is an important light textile industry production base and an international trade window province, and cross-border e-commerce has developed rapidly.

We increased the dialogue on emission reduction in transportation industry (line 560): Therefore, Fujian Province should attach importance to the research, development and production of new energy vehicles, and actively support the development of intelligent networked vehicles and hydrogen fuel cell vehicles in cities and regions with conditions, so as to reduce the carbon emissions of the transportation, storage, post and telecommunication services.

  • We have increased our view on the future decoupling of carbon emissions in the construction industry(line 605): In general, the industry is currently in a decoupling state, and the impact of various factors on the industry’s carbon emissions is small. In the future, the industry may be in a stable decoupling state for a long time.

  1. The third is about the modification of 4.3 (Analysis of Industry Carbon Decoupling Efforts in Fujian Province).

  • The first is the viewpoint that population growth will promote carbon emissions (line 621): However, inhibiting population growth will lead to population aging. Therefore, it is the key to advocate a simple, moderate, green and low-carbon lifestyle, actively pro-mote the establishment of green and low-carbon demonstration as required, and improve the national awareness of energy conservation and low-carbon.

  • We cite empirical literature to support our research results about industrial structure. The details are as follows (line 626): Moreover, the optimization of industrial structure is an effective measure to reduce carbon emissions, which has been proved by previous scholars, such as Zhang et al. [33].

  • We cite empirical literature to support our research results about energy intensity. The details are as follows (line 634): Energy intensity also showed positive decoupling effort in most industries, . The result is consistent with the previous studies for Fujian provinces [68].

  • We discussed the reasons why the energy structure affects carbon emissions(line 645): Because the carbon emission coefficients of different types of energy are different, increasing the use of clean energy such as renewable energy and natural gas can effectively reduce carbon emissions [69], so optimizing the energy structure is an effective emission reduction measure.

  • We added a dialogue on energy structure optimization(line 653): Therefore, it is crucial to promote the research and development of renewable energy and energy storage technologies such as wind energy, solar energy, biomass energy, geothermal energy, marine energy and hydrogen energy, and accelerate the realization of new breakthroughs in nuclear energy and nuclear safety technology, smart grid and building energy conservation technology to reduce carbon emission in Fujian Province.

We have marked all changes in the manuscript. Thanks for your time and patience again!

Round 3

Reviewer 2 Report

You worked hard to improve the paper.